# A morphological transformation in respiratory syncytial virus leads to enhanced complement deposition

Jessica P Kuppan[†], Margaret D Mitrovich[†], Michael D Vahey*

Department of Biomedical Engineering and Center for Science & Engineering of Living Systems (CSELS), Washington University in St. Louis, St. Louis, United States

**Abstract** The complement system is a critical host defense against infection, playing a protective role that can also enhance disease if dysregulated. Although many consequences of complement activation during viral infection are well established, mechanisms that determine the extent to which viruses activate complement remain elusive. Here, we investigate complement activation by human respiratory syncytial virus (RSV), a filamentous respiratory pathogen that causes significant morbidity and mortality. By engineering a strain of RSV harboring tags on the surface glycoproteins F and G, we are able to monitor opsonization of single RSV particles using fluorescence microscopy. These experiments reveal an antigenic hierarchy, where antibodies that bind toward the apex of F in either the pre- or postfusion conformation activate the classical pathway whereas other antibodies do not. Additionally, we identify an important role for virus morphology in complement activation: as viral filaments age, they undergo a morphological transformation which lowers the threshold for complement deposition through changes in surface curvature. Collectively, these results identify antigenic and biophysical characteristics of virus particles that contribute to the formation of viral immune complexes, and suggest models for how these factors may shape disease severity and adaptive immune responses to RSV.

**\*For correspondence:**
mvahey@wustl.edu

[†]These authors contributed equally to this work

**Competing interest:** The authors declare that no competing interests exist.

## Introduction

The complement system is a network of proteins that play a vital role in the innate and adaptive immune responses to pathogens including viruses (*O'Brien et al., 2011*; *Kopf et al., 2002*; *Bottermann et al., 2019*), bacteria (*Bladen et al., 1966*; *Mishra et al., 2012*), and parasites (*Roestenberg et al., 2007*; *Engstler et al., 2007*; *Kurtovic et al., 2018*). Activation of the complement system proceeds through three principal routes: the classical, lectin, and alternative pathways (reviewed in *Dunkelberger and Song, 2010*). These pathways differ in their mechanisms of activation: by opsonizing antibodies (classical pathway), by pathogen-specific carbohydrates (lectin pathway), or through continual low levels of attachment to surfaces (alternative pathway). Following activation, each pathway converges on C3, the central component of the complement cascade. C3 that has been cleaved by proteases or that has spontaneously hydrolyzed can covalently attach to activating surfaces via a reactive thioester. C3 attachment to surfaces is self-amplifying, producing new C3 convertases that further drive opsonization. C3 attachment also contributes to the terminal arm of the complement cascade, eventually leading to the assembly of a membrane attack complex that can neutralize membrane-bound targets through the formation of lytic pores.

In addition to its role in the neutralization of pathogens and infected cells, C3 also plays a central role in immune signaling. Upon attachment to pathogen surfaces, multiple C3 cleavage products interact with a variety of immune receptors. These interactions contribute to antigen transport to and within lymphoid organs (*Ochsenbein et al., 1999*; *Phan et al., 2007*); antigen presentation by follicular dendritic cells (*Reynes et al., 1985*); activation of B cells (*Hebell et al., 1991*; *Dempsey et al.,*

1996); T cell priming (*Kopf et al., 2002*); and the clearance of immune complexes by phagocytic cells (*Lukácsi et al., 2017*); or erythrocytes (*Cornacoff et al., 1983*). Additionally, the cleavage product C3a, also produced during complement activation, is a potent anaphylatoxin, increasing inflammation and recruiting immune cells to sites of infection (*Bera et al., 2011*; *Peng et al., 2009*). The diversity of interactions between immune cells and C3 highlights its central protective role bridging innate and adaptive immunity, as well as the potential dangers associated with dysregulation of complement (*Gralinski et al., 2018*; *Ricklin et al., 2016*). Although the disparate contributions complement makes to health and disease are well established, the mechanisms by which different human viruses activate or evade complement are less well understood. Understanding the factors that contribute to this process could help aid in the development of more effective vaccines and improve understanding of disease pathogenesis.

Here, we set out to investigate complement activation by the human pathogen respiratory syncytial virus (RSV). Activation of complement during RSV infection has been linked to both protection (*Corbeil et al., 1996*; *Bukreyev et al., 2012*) and pathogenesis (*Bera et al., 2011*; *Polack et al., 2002*), but the mechanisms that drive complement activation by RSV remain unclear. RSV is an enveloped, negative-sense single-stranded RNA virus of the family *Pneumoviridae* that causes severe infection among infants, the immunocompromised, and the elderly. The RSV genome encodes three membrane proteins – the fusion protein (F), the attachment glycoprotein (G), and the short hydrophobic protein (SH) – which are expressed on the surface of infected cells and packaged to varying degrees into shed virus particles. Among these surface proteins, F and G serve as the primary targets of the adaptive immune response, as well as the leading candidates for vaccine development and prophylaxis (*McLellan et al., 2013b*; *Fedechkin et al., 2018*; *Gilman et al., 2016*; *Collarini et al., 2009*). The fusion protein, F, mediates the merger between viral and cellular membranes during RSV entry (*Battles and McLellan, 2019*) and has recently been reported to induce outside-in signaling via IGF-1R to facilitate this process (*Griffiths et al., 2020*). The glycoprotein, G, mediates attachment through the chemokine receptor CX3CR1 (*Johnson et al., 2015*; *Chirkova et al., 2015*; *Zhivaki et al., 2017*) and, when expressed in soluble form, helps antagonize immune responses (*Bukreyev et al., 2012*). While antibodies against F can provide potent protection both in vitro and in vivo, the conformational rearrangements this protein undergoes have historically presented challenges in vaccine design (*Graham, 2017*). Although antibodies against the prefusion conformation of F (pre-F) are frequently capable of blocking viral entry, antibodies that bind to the postfusion conformation of F (post-F) often fail to do so (*Goodwin et al., 2018*). Numerous recent breakthroughs in protein design are helping to overcome this challenge through the development of stabilized pre-F antigens as vaccine candidates (*McLellan et al., 2013a*; *Marcandalli et al., 2019*; *Crank et al., 2019*). However, in the context of natural infection, both pre- and post-F conformations occur, and high antibody titers against post-F have been associated with enhanced disease severity and increased activation of complement in the lungs (*Polack et al., 2002*; *Acosta et al., 2015*).

Understanding how antibodies and RSV antigens contribute to complement activation could provide new insights into vaccine development and mechanisms of RSV pathogenesis. To investigate how RSV-specific antibodies contribute to activation of complement, we developed a fluorescence imaging-based approach to simultaneously quantify antibody binding, the abundance of viral antigens, and the deposition of complement proteins on RSV at the single-virus level. These experiments identify an antigenic hierarchy for complement deposition, where antibodies that bind to some antigenic sites activate the classical pathway while other antibodies do not. This hierarchy is dictated by accessibility of the antibody Fc for binding by C1, the initiator of the classical pathway. We also identify a role for the complement defense protein CD55 (DAF), which is packaged into virus particles and restricts C3 deposition. Finally, we identify biophysical features of individual RSV particles within heterogeneous populations that enhance their tendency to drive complement deposition. In particular, we find that the detachment of the RSV matrix from the viral envelope – a transition that can be induced by physical perturbations but also occurs under normal physiological conditions in vitro – significantly enhances complement deposition by a range of antibodies targeting either pre-F, post-F, or both conformations of F. Collectively, these results identify a constellation of mechanisms that contribute to complement activation and immune complex formation by RSV, and inform models for how complement may shape disease severity and the adaptive immune response to infection.

## Results

### A fluorescence-based approach to measuring complement deposition on RSV particles

To identify determinants of complement activation by RSV, we developed a fluorescence assay to quantify antibody binding and opsonization with complement C3 across populations of individual virus particles. Following our previous work using influenza A virus (*Vahey and Fletcher, 2019a*), we engineered a strain of RSV A that was amenable to fluorescence imaging of infected cells and shed virus particles (*Supplementary file 1*, *Figure 1—figure supplement 1A*). To track viral infection, we inserted a fluorescent reporter (mTagBFP2) downstream of the NS1 stop codon but upstream of the gene-end sequence using an internal ribosome entry site (IRES). Separately, we inserted a ybbR-tag at the C-terminus of G and a pentaglycine tag immediately following the signal sequence in F. Sfp synthase (*Yin et al., 2006*) can be used to conjugate CoA-based fluorescent probes to the tag on G, while the tag on F becomes exposed following cleavage of the signal sequence, creating a suitable substrate for the enzyme Sortase A (SrtA) to conjugate small peptide-based fluorophores (*Theile et al., 2013*). These modifications allow us to visualize RSV particles and measure abundance of antigen (i.e. F or G) on the viral surface independent of conformational changes in F (i.e. prefusion vs. postfusion) and while preserving antigenic sites. Viruses labeled in this way retain levels of infectivity matching unlabeled controls (*Figure 1—figure supplement 1B*), demonstrating that the attachment of fluorophores is non-disruptive. Additional characterization of this labeling approach and our image acquisition settings verified that intensity measurements are linear with fluorophore concentration on RSV particles (*Figure 1—figure supplement 2*), and that a high percentage (~90%) of F on the surface of RSV particles is labeled (*Figure 1—figure supplement 3*) (see also Materials and methods).

Using this system, we sought to characterize activation of the classical pathway by RSV particles. We enzymatically labeled F on the surface of RSV particles collected from A549 cells and immobilized these viruses onto PEGylated coverslips functionalized with the anti-G antibody 3D3 (*Fedechkin et al., 2018*). This allowed us to quantify opsonization following incubation of fluorescent viruses with normal human serum (NHS) supplemented with defined fluorescent monoclonal antibodies (mAbs) and labeled C3 (*Figure 1A*). Initial tests using NHS resulted in robust C3 deposition in the absence of supplemental mAbs (*Figure 1—figure supplement 4A*, left column). Incubating virus with this serum for 30 min at 4 °C also inhibited binding of a high-affinity pre-F-specific mAb (5C4) ~ 2 -fold and a post-F-specific mAb (ADI-14359) by ~10 -fold, suggesting the presence of competing polyclonal IgG/IgM within the serum (*Figure 1—figure supplement 4B*). In contrast, IgG/IgM-depleted NHS showed little C3 deposition in the absence of supplemental mAbs, but robust opsonization with C3 when F-specific mAbs were added (*Figure 1—figure supplement 4A*, right column). IgG/IgM-depleted serum therefore provides a means of determining the ability of individual mAbs to drive complement deposition, allowing us to identify antibody features that are predictive of potency.

### Complement deposition driven by F-specific antibodies varies by antigenic site

We expressed and purified a panel of human IgG1 mAbs targeting six antigenic sites of RSV F (*Gilman et al., 2016*; *Figure 1B*). Each mAb was modified to contain a ybbR-tag (*Yin et al., 2006*) at the C-terminus of the heavy chain, to permit quantitative site-specific labeling and the ability to measure the amounts of antibody bound to individual virus particles. For these experiments, we use mAbs at concentrations similar to those measured for dominant clonotypes in post-vaccination serum (*Lavinder et al., 2014*) (10–20 µg/ml), well above the equilibrium-binding affinity of these antibodies. By using mAbs at concentrations that saturate binding, we were able to separate the intrinsic capacity of a given mAb to drive complement deposition when bound to antigen from that mAb's binding affinity. At saturated binding, these mAbs vary markedly in their ability to activate the classical pathway. Among the pre-F-binding mAbs tested, 5C4 (*McLellan et al., 2013b*) (site 0), CR9501 (*Gilman et al., 2019*) (site V), and Motavizumab (*Wu et al., 2007a*) (site II) showed the greatest potency. In contrast, ADI-19425 (*Goodwin et al., 2018*; *Wu et al., 2007a*) (site III) and 101F (*Wu et al., 2007b*) (site IV) showed only background levels of C3 deposition (*Figure 1C & D*). Thus, although each pre-F-binding mAb achieved similar levels of binding to RSV F (*Figure 1D*, lower plot), only three out of five led to C3 deposition above background levels.

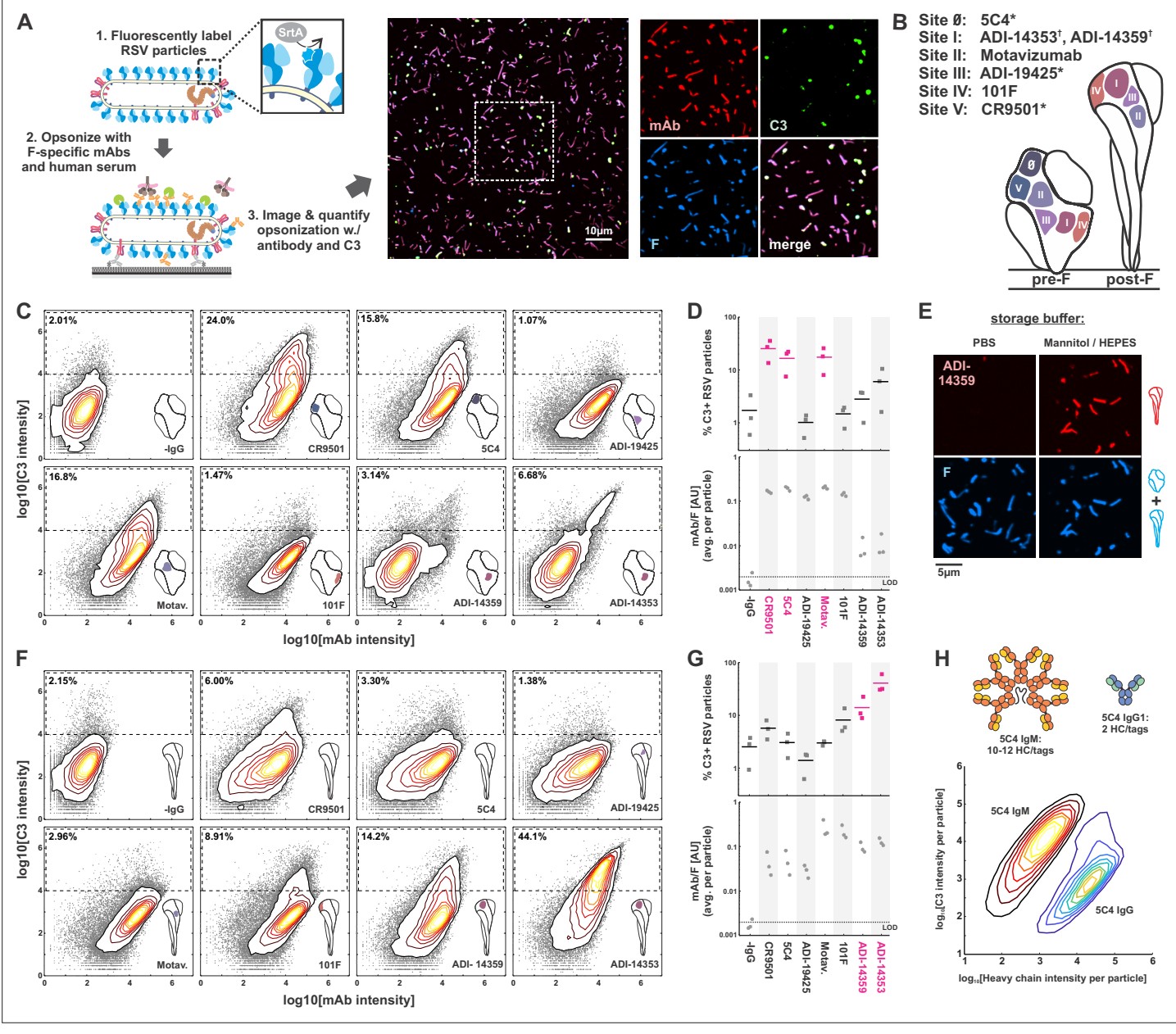

**Figure 1.** Complement activation and C3 deposition vary across antigenic sites of RSV F. (**A**) A fluorescence-based approach to measuring opsonization of RSV particles with mAbs and C3. RSV particles with site-specifically labeled F are immobilized on coverslips and incubated with normal human serum (IgG/IgM-depleted), specific mAbs, and fluorescent C3 prior to imaging. *Right*: RSV particles opsonized with mAb (CR9501 IgG1 in the image shown) and C3. (**B**) Antibodies used in this study and their antigenic sites. A '*' denotes mAbs specific to prefusion F while a '†' denotes mAbs specific to postfusion F. (**C**) Distributions of integrated antibody and C3 intensities on opsonized virus particles with predominantly prefusion F. Gray points indicate data for individual virus particles. Dashed regions indicate criteria for C3-positive particles. Data is combined from three biological replicates. (**D**) *Top*: Data from C, plotted as percentage of C3-positive RSV particles, defined by integrated C3 intensities > $10^4$. Points show results from three biological replicates with the mean across the replicates shown as a line. Antibodies determined to be activators of the classical pathway (>10% C3-positive particles) are shown in magenta. *Bottom:* Plot of average mAb:F intensity per RSV particle across the same antibodies and replicates as in the plots above. (**E**) Conversion of pre-F to post-F on RSV filaments via ~24 hr incubation in buffer with low ionic strength but balanced osmolarity. Post-F is detected using the site I-directed mAb ADI-14359. Images are displayed at matching contrast levels. (**F**) and (**G**): Results corresponding to C and D but for RSV particles containing predominantly post-F. (**H**) Comparison of C3 deposition by IgM and IgG1 antibodies. *Top*: Schematic of antibodies based on 5C4. The antibodies contain matching light chains and VH domains coupled to the human heavy chain μ (for IgM), or a human IgG1 Fc. IgM is additionally co-expressed with a J chain plasmid. All heavy chains contain a c-terminal ybbR tag for site-specific conjugation of a fluorophore for quantification of bound antibody. *Bottom*: Contour plots showing distributions of C3 and IgM/IgG1 heavy chains per RSV particle, based on fluorescence intensities. The IgM distribution is determined from 317594 RSV particles; the IgG distribution is determined from 126372 RSV particles.

*Figure 1 continued on next page*

*Figure 1 continued*

See also *Figure 1—source data 1*.

The online version of this article includes the following figure supplement(s) for figure 1:

**Source data 1.** Matlab code and data for *Figure 1*, *Figure 1—figure supplement 1*, *Figure 1—figure supplement 2*, *Figure 1—figure supplement 3*, and *Figure 1—figure supplement 4*.

**Figure supplement 1.** A fluorescence imaging-based approach to study complement deposition on RSV.

**Figure supplement 2.** Determining the linearity of fluorescence measurements.

**Figure supplement 3.** Estimating the efficiency of enzymatic labeling.

**Figure supplement 4.** Effects of IgG/IgM depletion on complement deposition and binding of RSV-specific mAbs.

Several F-specific antibodies bind to the postfusion conformation of F, either exclusively or in addition to the prefusion conformation. This includes several antibodies in our panel: 101 F and Motavizumab (which bind to both pre- and post-F) as well as ADI-14359 and ADI-14353 (which bind specifically to post-F) (*Goodwin et al., 2018*). To compare C3 deposition driven by these antibodies upon binding to post-F, we first incubated viruses bound to coverslips for 24 hr in buffer with low ionic strength (300 mM mannitol, 10 mM HEPES pH 7.2), based on the previous finding that pre-F spontaneously converts to post-F in buffers with low salt concentrations (*Hicks et al., 2018*; *Chai-watpongsakorn et al., 2011*). This treatment increased binding of the post-F-specific antibodies ADI-14359 and ADI-14353 ~300-fold without obvious changes in the morphology of virus particles (*Figure 1E*). Conversion of RSV particles to a predominantly post-F form increased C3 deposition from post-F-specific mAbs ADI-14359 and ADI-14353 to levels comparable to or greater than the most potent pre-F-specific antibodies in the context of pre-F antigens (*Figure 1F & G*). For the conformation-independent mAbs 101 F and Motavizumab, C3 deposition following pre-to-post-F conversion followed different trends, increasing ~6-fold for 101 F but decreasing by a similar ratio for Motavizumab, despite similar levels of antibody binding (*Figure 1F & G*).

The classical pathway is activated when the C1 initiation complex binds to IgM or IgG that has assembled on activating surfaces. While activation via IgG requires the assembly of a hexameric complex of antibody Fc regions (*Diebolder et al., 2014*; *Ugurlar et al., 2018*; *Wang et al., 2016*), IgM is pre-assembled as an activating platform, with C1-binding sites exposed only upon engagement with surface antigen (*Sharp et al., 2019*). As a comparison with our IgG antibodies, we tested activation of the classical pathway by recombinant IgM with VH and VL domains from 5C4. As expected, this high-affinity IgM was considerably more potent than its IgG1 counterpart, leading to >100 -fold more C3 deposition per bound heavy chain than the corresponding IgG1 mAb (*Figure 1H*). Given the importance of establishing a platform for C1 binding in activation of the classical pathway, we sought to identify how the IgG antibodies in our panel may differ in this regard.

A consistent trend among the IgG1 antibodies that most efficiently drive complement deposition is the angle with which they bind to either pre- or post-F. 101 F and Motavizumab, antibodies that bind to both conformations but with alternating Fc orientations, illustrate this effect. For both antibodies, C3 deposition increases ~6-fold when the Fc is oriented away from the viral membrane as opposed to toward it. Projection of the Fc region further above the viral membrane is common to all of the activating antibodies we tested (*Figure 2A*). This may increase accessibility for binding by C1, and positioning the Fc regions on a plane above the surrounding canopy of F could also facilitate Fc hexamer formation (*Diebolder et al., 2014*) by avoiding steric hindrance from neighboring proteins in the viral membrane. Consistent with this model, we observe more binding by C1 (as detected by an anti-C1q antibody) to mAbs that bind to pre-F with their Fcs projected outward (CR9501, 5C4, Motavizumab) compared to those where the Fc lies within or below the plane of F trimers in the viral membrane (ADI-19425, 101 F) (*Figure 2B & C*). Of note, we still observe C1 binding for mAbs that do not drive C3 deposition (e.g. ADI-19425), suggesting that attachment of C1 to these mAbs may be less likely to produce an active C1 complex. This could occur if a high density of antibodies permitted C1 to attach, but assembly of the activating hexamer was occluded by F, G, or other proteins present at high densities on the viral surface.

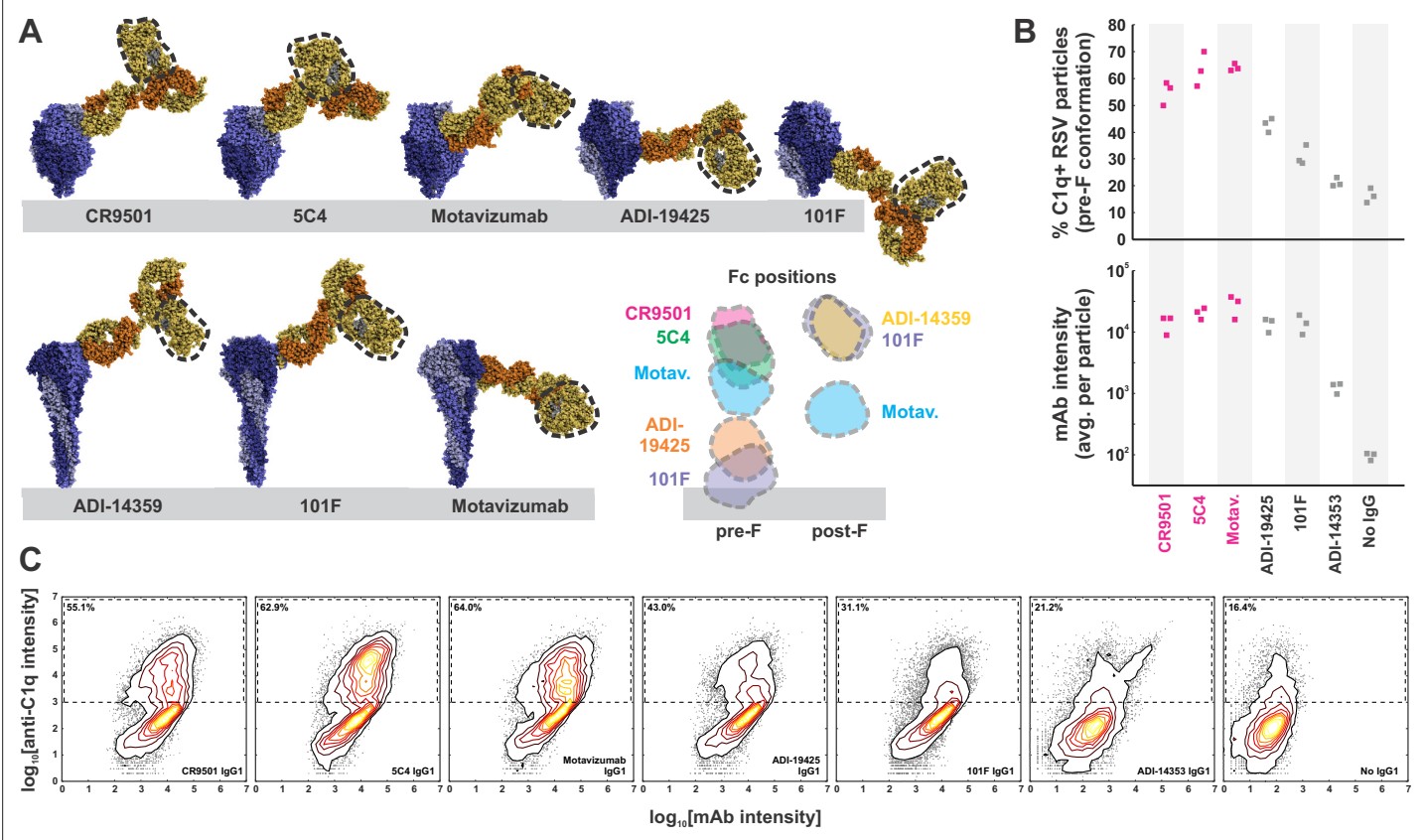

**Figure 2.** Complement activation and C1q binding varies with Fc position. (**A**) Modeling Fc positions for F-specific mAbs. Structures for RSV F (or portions thereof) with bound antibodies (PDB IDs 6OE4, 5W23, 4ZYP, 3IXT, 6APD, 6APB, and 3O45) were aligned with human IgG1 (PDB ID 1HZH) to determine representative locations accessible to antibody Fc regions (indicated by dashed outline). Distances from the viral membrane range from ~1 nm (101 F) to ~18 nm (CR9501, ADI-14359). (**B**) C1q binding to predominantly pre-F RSV particles opsonized with different antibodies. The top plot shows the percentage of C1q+ particles, defined as those with a total intensity of anti-C1qA antibody >$10^3$. The bottom plot shows the intensity of anti-F mAb for each condition. Individual points represent values for three biological replicates. Antibodies determined to activate complement from pre-F antigens (*Figure 1*) are shown in magenta. (**C**) Distributions of anti-F mAb intensities (horizontal axis) and anti-C1qA antibody intensities (vertical axis) for different anti-F mAbs bound to pre-F particles. Particles within the dashed rectangles indicate those that are C1q-positive, and the percentage of these particles is indicated in the upper left. Distributions are combined from the same three biological replicates represented in B. See also *Figure 2—source data 1*.

The online version of this article includes the following figure supplement(s) for figure 2:

**Source data 1.** Matlab code and data for *Figure 2*.

## CD55 is packaged into RSV particles and modulates sensitivity to complement deposition

Complement defense proteins anchored in the membranes of host cells can be packaged into enveloped viruses during assembly (*Li and Parks, 2018*; *Hutchinson et al., 2014*; *Marschang et al., 1995*), where they may function to restrict different stages of the complement cascade. To determine if host complement defense proteins restrict opsonization of RSV particles with C3, we focused on the roles of CD46 and CD55. Both proteins are abundantly expressed on A549 cells and function to limit the amplification step of the complement cascade by restricting the formation or stability of new C3 convertases (*Noris and Remuzzi, 2013*). Using fluorescent Fab fragments or antibodies against CD46 and CD55, we were able to detect both on the surface of RSV particles collected from A549 cells. We proceeded to construct two polyclonal A549 knockout (KO) cell lines where one gene or the other was deleted via CRISPR/Cas9. Following KO and cell sorting, we were no longer able to detect the targeted proteins in cells or shed viruses, verifying the specificity of the antibodies and confirming successful KO (*Figure 3A&B*). Using these cells lines, we proceeded to compare C3 deposition in the presence or absence of CD55 or CD46. Virus released from CD55 KO cells showed increased

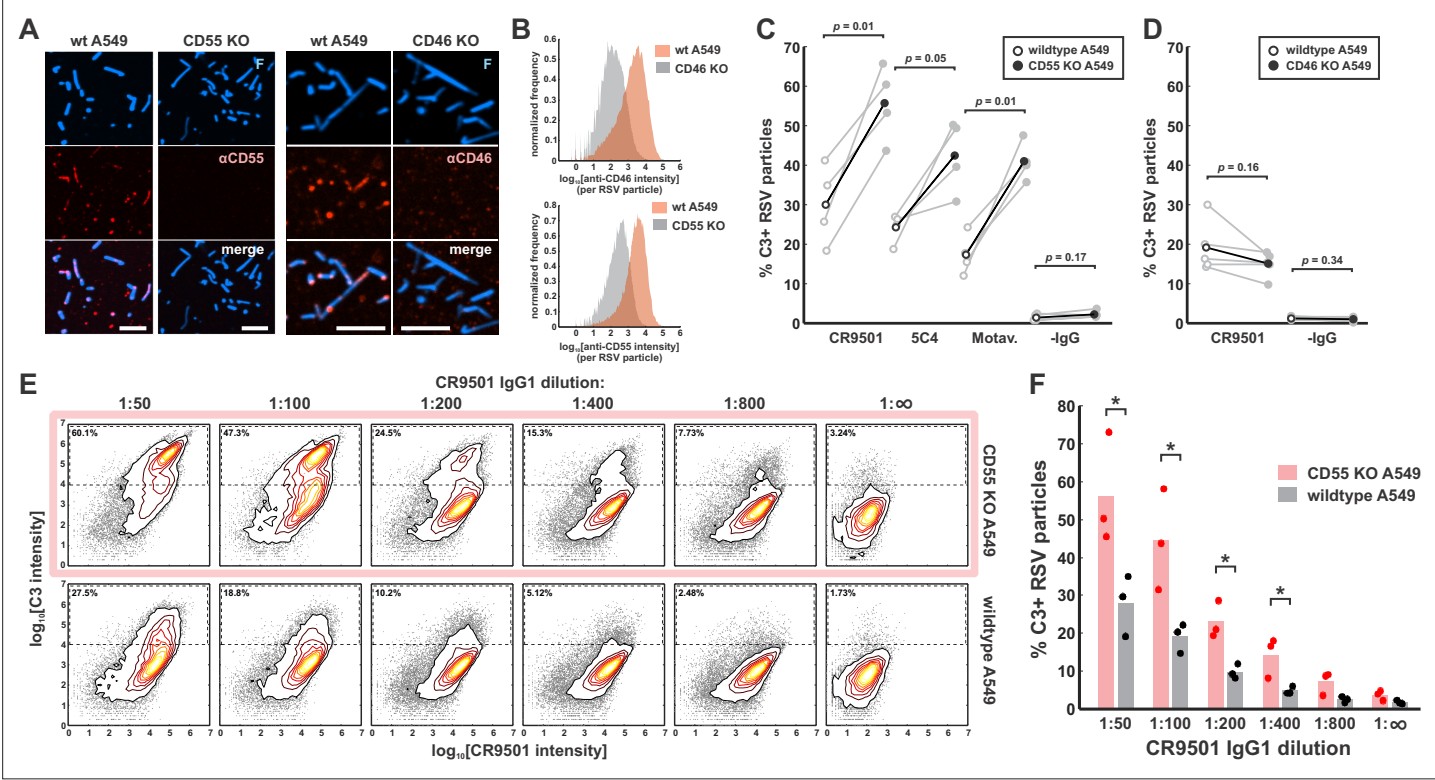

**Figure 3.** CD55 is packaged into RSV particles and increases thresholds for C3 opsonization. (**A**) Images of RSV particles with enzymatically labeled F and CD55 or CD46 labeled via fluorescent antibodies. Panels show representative images of virus released from wildtype (wt) A549 cells and CD55 or CD46 knockout (KO) cells, displayed at matching contrast levels. Scale bar = 5 µm. (**B**) Histograms showing distributions of antibody intensities per RSV particle for wt and KO cell lines. (**C**) Comparison of C3 deposition on viruses raised in wt cells (open circles) or CD55 KO cells (closed circles) using three different F-specific antibodies and a negative control. Black markers show average values across four biological replicates. Individual replicates are shown in gray. Connecting lines indicate samples that were prepared in parallel, using virus collected from separate batches of wt and CD55 KO cells. p-Values are determined using a paired-sample t-test. (**D**) Similar plot as in C, but for data obtained from CD46 KO cells. (**E**) Distributions of antibody and C3 intensities for serial dilutions of CR9501 IgG1. Panels in the top row show results for viruses from CD55 KO cells, while panels from the bottom row show results for viruses from wt cells. The region indicated by the dashed line represents the threshold for C3-positive particles. Percentages in the upper left corners indicate the percentage of C3-postiive particles. Data is combined from three biological replicates. (**F**) Data from E plotted as percentage C3-positive particles per condition. Bars indicate mean values and points show individual replicates. * indicates p < 0.05 calculated using a two-sample t-test. See also *Figure 3—source data 1*.

The online version of this article includes the following figure supplement(s) for figure 3:

**Source data 1.** Matlab code and data for *Figure 3*.

sensitivity to C3 deposition across antibodies specific to sites II and V, with percentages of opsonized particles increasing ~2- to 3-fold relative to virus produced by wildtype cells (*Figure 3C*). Conversely, virus released from CD46 KO cells did not show significant differences in C3 deposition relative to wildtype cells using the site-V-specific mAb CR9501 (*Figure 3D*). Comparing activation by CR9501 across a range of antibody dilutions suggests that deletion of CD55 increases opsonization to an extent comparable to a 4-fold increase in antibody concentration (*Figure 3E&F*). While additional complement defense proteins may play critical roles in other aspects of RSV infection, these results show that CD55 plays an outsized role in modulating sensitivity of shed virus particles to opsonization with C3.

## Globular RSV particles serve as dominant targets of complement deposition

Although the efficiency of opsonization with C3 varies depending on the activating antibody and on the presence or absence of host complement defense proteins, we observed a consistent pattern across experimental conditions, where particles opsonized with C3 had more globular morphology

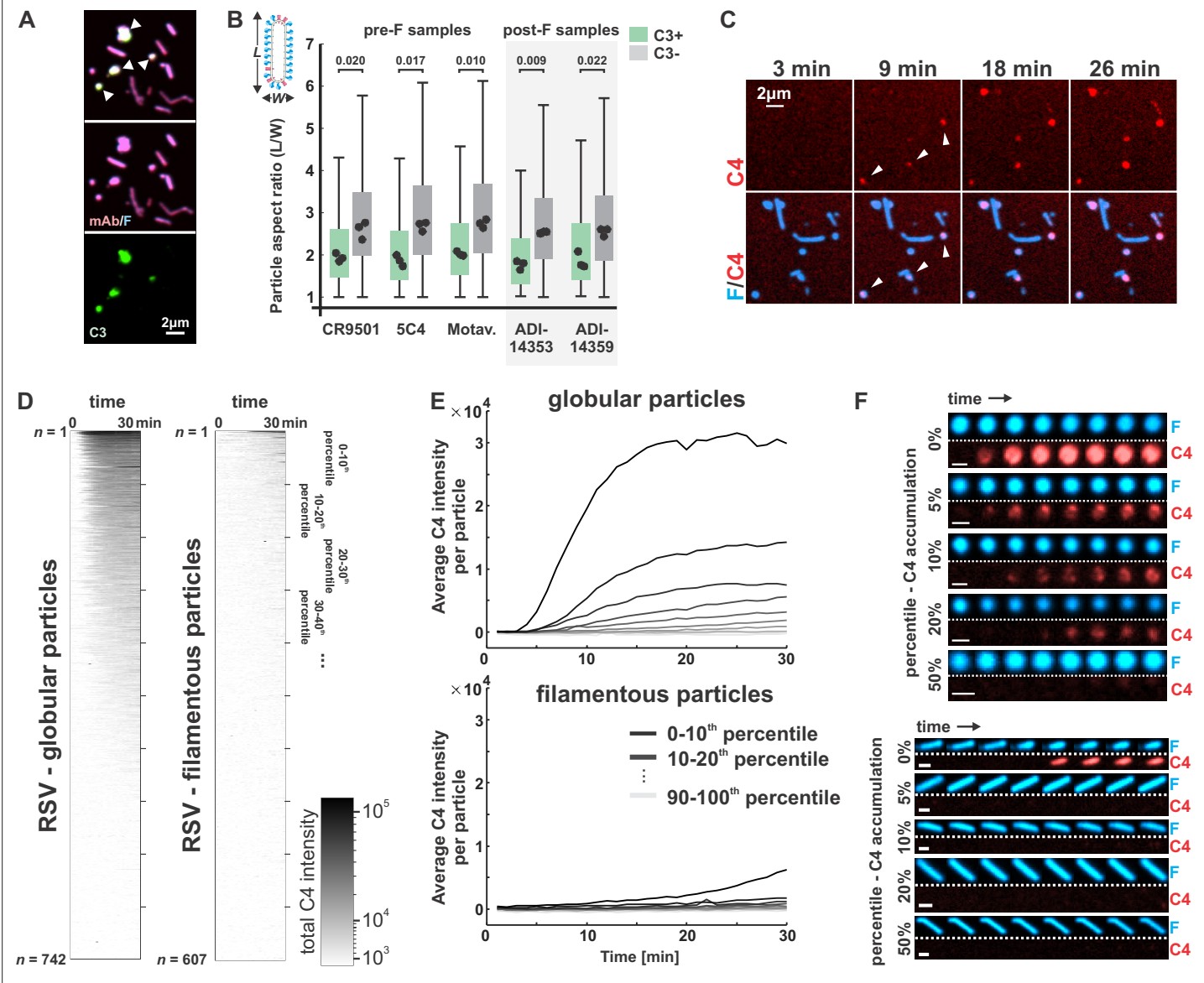

**Figure 4.** Globular particles are preferentially and rapidly opsonized with C3 and C4. (**A**) RSV particles opsonized with antibody (CR9501 IgG1) and C3. White arrows in merged panel indicate globular particles with high levels of C3. (**B**) Comparison of particle aspect ratios across RSV particles that do (green boxes) or do not (gray boxes) show opsonization with C3. Boxes indicate 25th–75th percentiles and points show median values for three biological replicates. Analysis is limited to particles with an area larger than 100 pixels, where morphology can be determined from diffraction-limited images. p-Values are determined based on median values using a paired sample t-test. (**C**) Time series of C4 accumulation on RSV particles. White arrows indicate first detectable accumulation of C4. (**D**) C4 accumulation on RSV particles over time, categorized by particle morphology. Data to the left shows C4 accumulation on 742 globular particles; data on the right show C4 accumulation for 607 filamentous particles. In both plots, rows show C4 data for individual particles from 0 to 30 min, sorted from top to bottom according to total C4 accumulation. (**E**) Data from D, plotted by grouping particles within percentile intervals from 0% to 10%, 10% to 20%, etc. and averaging C4 accumulation in each group. (**F**) Sample images of globular (top) and filamentous (bottom) RSV particles. Displayed images are sampled at 4 min intervals beginning from 2 min after the addition of serum and complement components (scale bar = 1 μm). See also *Figure 4—source data 1*.

The online version of this article includes the following figure supplement(s) for figure 4:

**Source data 1.** Matlab code and data for *Figure 4*.

than those with low or undetectable levels of C3, which tended to be more filamentous (*Figure 4A & B*).

The bias in C3 opsonization toward globular particles could reflect a higher intrinsic sensitivity to complement activation in these particles as compared to viral filaments, or it could indicate that

particle morphology is altered upon deposition of C3 or other complement proteins. To determine if morphological differences precede or follow from differences in complement deposition, we monitored deposition of C4 on RSV particles over time, using CR9501 as the activating antibody. The lower concentrations of C4 in serum as compared to C3 allowed us to visualize deposition on particles without the high background signal from protein in solution that arises when using C3. Additionally, deposition of C4b is immediately downstream of C1 in the classical pathway, providing rapid detection of activation. Consistent with our endpoint observations of C3 deposition, C4 accumulates on globular particles first, appearing within ~10–15 min of incubation with complement components (*Figure 4C*). Analysis of C4 deposition across n = 742 globular and n = 607 filamentous particles revealed substantial differences in patterns of opsonization; ~50 % of globular particles showed detectable accumulation of C4 within 30 min, compared to <10% of filamentous particles (*Figure 4D–F*). Moreover, opsonization with C4 proceeds with different kinetics, occurring ~5 -fold faster in globular particles than in filamentous ones (*Figure 4E&F*). These results demonstrate that sensitivity to antibody-dependent complement deposition correlates with RSV particle morphology.

## Globular particles arise spontaneously over time and are enriched in post-F, but do not require post-F mAbs to drive complement deposition

We next sought to determine how the epitopes presented on the surface of globular particles differ from their filamentous counterparts. Prior studies using electron microscopy have reported that globular RSV particles are enriched in post-F (*Liljeroos et al., 2013*). We characterized particle morphology and F conformation by simultaneously labeling RSV particles with the prefusion-specific antibody 5C4 and the postfusion-specific antibody ADI-14359. This revealed that filamentous particles contained almost exclusively pre-F while globular particles were frequently enriched in post-F (*Figure 5A*), corroborating previous work. To determine if these changes were the result of aging, we compared newly released virus with virus incubated in cell culture media at 37 °C for a total of 24 hr. These experiments revealed that the proportion of post-F-containing particles increased following the 24 hr incubation, along with the proportion of globular particles (*Figure 5B–D*). Direct labeling of RSV-infected cells with 5C4 and ADI-14359 revealed mixtures of pre- and post-F particles on the surfaces of both infected A549 cells and on neighboring uninfected cells, confirming that the occurrence of both particle types is not limited to circumstances where the virus is being collected or handled (*Figure 5—figure supplement 1*). Collectively, these results suggest that RSV particles transform spontaneously over time under mild conditions into a globular state, and that these globular particles are frequently enriched in post-F.

We proceeded to compare C3 deposition on particles that were newly shed from cells to those that had been aged 24 hr. We observed significant increases in C3 deposition on aged particles relative to new ones, regardless of whether a post-F-specific mAb (ADI-14359) or pre-F-specific mAb (CR9501) was used (*Figure 5E & F*). Thus, even particles with lower levels of pre-F showed an increased tendency to activate the classical pathway via pre-F-specific mAbs after aging. This result was surprising, suggesting that some feature of aged particles must be lowering thresholds for activation of the classical pathway. We therefore sought to determine mechanistically how differences in particle morphology arise, and how these differences could lower thresholds for complement deposition even when there are fewer epitopes available that are recognized by the activating antibody.

## Detachment of the viral matrix drives changes in RSV shape

The globular RSV particles we observe resemble prior observations from electron microscopy, where the matrix protein appears to have dissociated from the membrane, resulting in more rounded morphology and less ordered distributions of surface proteins as compared to viral filaments budding from infected cells (*Liljeroos et al., 2013*; *Ke et al., 2018a*). Although this effect has been attributed to damage during sample preparation, in our experiments this change occurs naturally as particles age, in the absence of harsh treatments. We reasoned that a similar effect could be achieved through osmotic swelling, allowing us to alter virus shape while better preserving pre-F on the virus surface. Treatment of RSV particles with a low osmolarity buffer (10 mM HEPES pH 7.2, 2 mM $CaCl_2$) transformed viral filaments into spherical particles over the course of ~1 min, with no loss of infectivity (*Figure 6A*, *Figure 6—figure supplement 1A*). Osmotic swelling also led to no loss in pre-F shortly

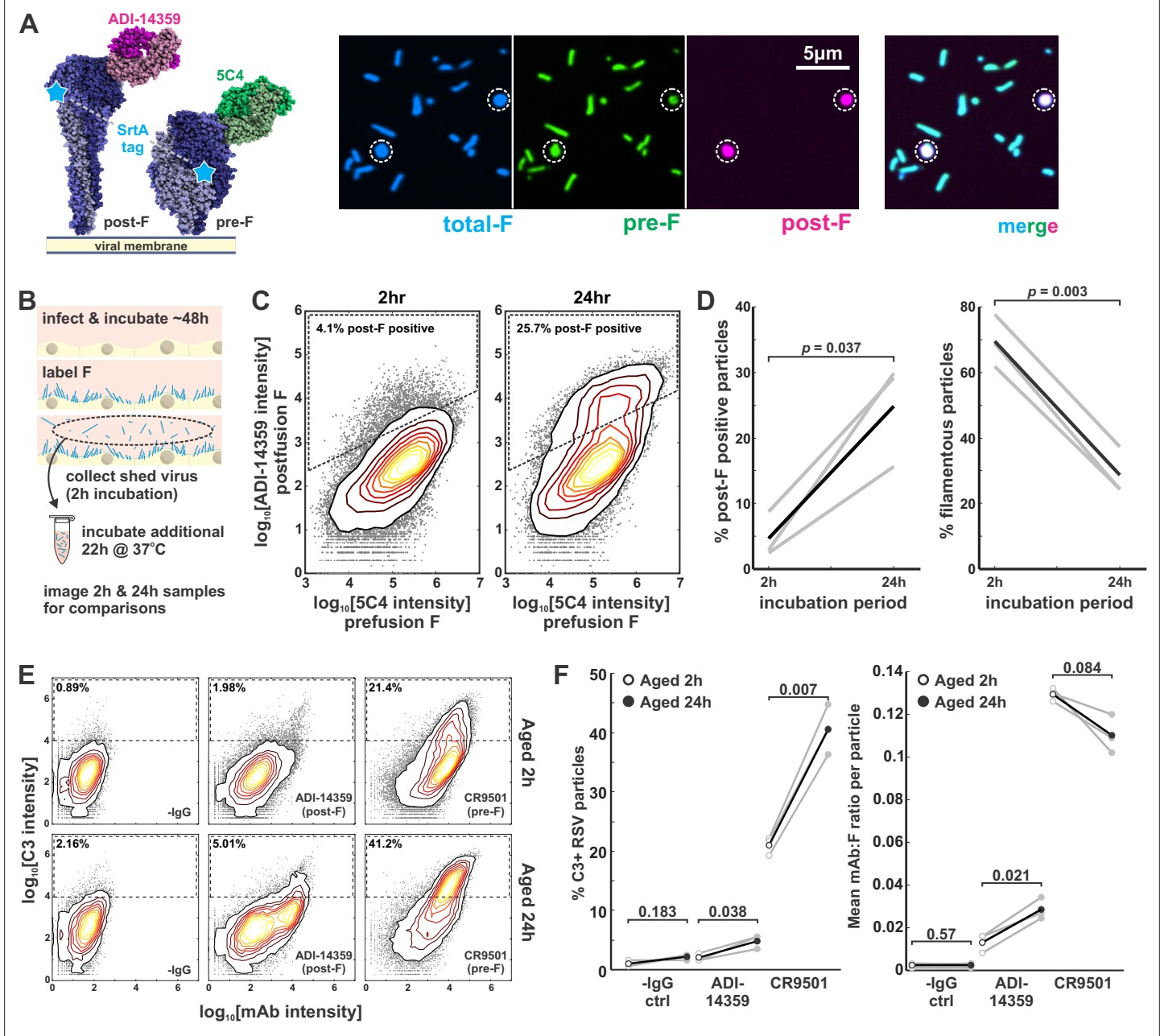

**Figure 5.** Globular particles are enriched in post-F, but remain sensitive to pre-F-mediated complement deposition. (**A**) Three-color labeling strategy to detect total F (via enzymatic labeling with Sortase A [SrtA]), post-F (via the post-F-specific mAb ADI-14359), and pre-F (via the pre-F-specific mAb 5C4). Fluorescence images show RSV particles labeled to indicate pre-F, post-F, and total F on the virion surface. (**B**) Experimental approach to determine effects of aging on RSV particles. (**C**) Distributions of pre-F and post-F intensities for virus aged 2 hr at 37 °C (left) or 24 hr at 37 °C (right). Data is combined from three biological replicates. Region inside the dashed lines defines criteria for post-F-positive particles. (**D**) Percentage of post-F-positive particles (left) and filamentous particles (right) after 2 and 24 hr aging. Post-F-positive particles are defined as those within the dashed lines in C. Filamentous particles are defined as those with length >1 µm and aspect ratio (L/W) > 2. Gray lines show results from paired biological replicates; black lines show average values. p-Values are determined using a paired sample t-test. (**E**) Distributions of C3 and mAb intensities per RSV particle in the absence of mAb ('-IgG') or in the presence of ADI-14359 or CR9501. The top row shows results for particles aged for a total duration of 2 hr at 37 °C (i.e. during budding from A549 cells); the bottom row shows particles aged for a total duration of 24 hr at 37 °C following collection at 2 hr. Particles within the dashed rectangles indicate those that are C3-positive, and the percentage of these particles is indicated in the upper left. Distributions are combined from three biological replicates. (**F**) *Left*: Percentage of C3-positive particles following aging at 37 °C for 2 hr (open circles) or 24 hr (filled circles). Black markers give average values for three biological replicates; individual replicates are shown in gray. *Right*: Mean mAb/F intensities per particle for the same datasets plotted to the left. p-Values determined using a paired-sample t-test.     See also *Figure 5—source data 1*.

*Figure 5 continued on next page*

*Figure 5 continued*

The online version of this article includes the following figure supplement(s) for figure 5:

**Source data 1.** Matlab code and data for *Figure 5*.

**Figure supplement 1.** Pre-F and post-F containing RSV particles occur naturally in cell culture.

after treatment; however, after a 24 hr incubation at 37 °C, the ratio of post-F to pre-F was ~3.5 -fold higher in osmotically swollen particles than in those where filamentous morphology was preserved (*Figure 6—figure supplement 1B*), suggesting that interactions between the viral matrix and the cytoplasmic tail of F may help to stabilize F in its prefusion conformation.

To corroborate that this morphological transformation coincides with detachment of the viral matrix, we performed photobleaching experiments on RSV particles with enzymatically labeled F. In the absence of an intact matrix layer, we reasoned that F (which may interact with the matrix protein via its cytoplasmic tail; *Förster et al., 2015*; *Shaikh et al., 2012*) could freely diffuse laterally within the plane of the viral membrane and should therefore show increased recovery after partial photobleaching (*Figure 6—figure supplement 1C and D*). Consistent with this prediction, we observed significantly more recovery in bleached F on globular or osmotically swollen viruses compared to viral filaments. The increased recovery on osmotically swollen viruses could be reversed by treatment with 0.5 % PFA while preserving the particles' spherical geometry (*Figure 6—figure supplement 1D*). These results suggest that the morphological transformation we observe as RSV particles age or are subjected to physical perturbations is driven by the detachment of the matrix from the viral membrane.

## Detachment of the viral matrix increases complement deposition through changes in surface curvature

We next sought to determine the effects of this induced morphological transformation on interactions with complement proteins. We found that osmotic swelling led to increased C1 binding in the presence of F-specific mAbs (CR9501, 5C4, ADI-19425), but not in their absence (*Figure 6B*, left), indicating that the effect is not a non-specific consequence of swelling. Furthermore, increased C1 binding did not result from increased mAb binding, which remains constant or decreases slightly upon swelling (*Figure 6B*, right). Similar to the effects of aging on virus particles (*Figure 5*), C3 opsonization also increases upon osmotic swelling, an effect that is conserved across mAbs targeting a range of antigenic sites on pre-F and/or post-F (*Figure 6C*). Recombinant IgM antibody with VH and VL domains from 5C4 shows a similar effect, with C3 deposition increasing by ~50 % following osmotic swelling across a range of IgM dilutions (*Figure 6D*). Collectively, these results demonstrate that detachment of the viral matrix enhances complement deposition by both IgG and IgM antibodies, and that this enhancement arises early in the pathway, with the attachment of C1 to the viral surface.

Changes in the shape of RSV particles are accompanied by two notable biophysical differences: increased mobility of F in the membrane and decreased membrane curvature. While either could potentially contribute to enhanced complement deposition, we observed a similar fold-increase in C3 deposition on both fixed (0.5 % PFA) and unfixed viruses in the spherical state vs. the filamentous state (*Figure 6—figure supplement 1E*), suggesting the antigen mobility alone does not account for preferential opsonization of globular RSV particles. We therefore sought to determine if membrane curvature could be a contributing factor. Curvature has previously been implicated in complement activation on antigen/antibody-coated beads and peptidoglycan nanoparticles (*Pedersen et al., 2010*; *Zeuthen et al., 2020*; *Westas Janco et al., 2018*). However, these studies have reached different conclusions regarding the effects of curvature, and it remains unclear how these results would generalize to enveloped viruses, where antigens are oriented and may be spaced semi-regularly (*Wasilewski et al., 2012*; *Peukes et al., 2020*; *Ke et al., 2018b*).

The transformation from filament to sphere results in decreased membrane curvature to an extent that varies depending on the size of the initial filament. While the change in curvature is negligible for particles whose length and diameter are similar (~100 nm), curvature will decrease ~2 -fold for a particle 1 μm in length, and ~5 -fold for particles 10 μm in length (*Figure 6—figure supplement 2A and B*). If complement activation by IgG is sensitive to curvature, the effects of osmotic swelling on C3 deposition should therefore vary depending on the initial size of the particle. To test this prediction,

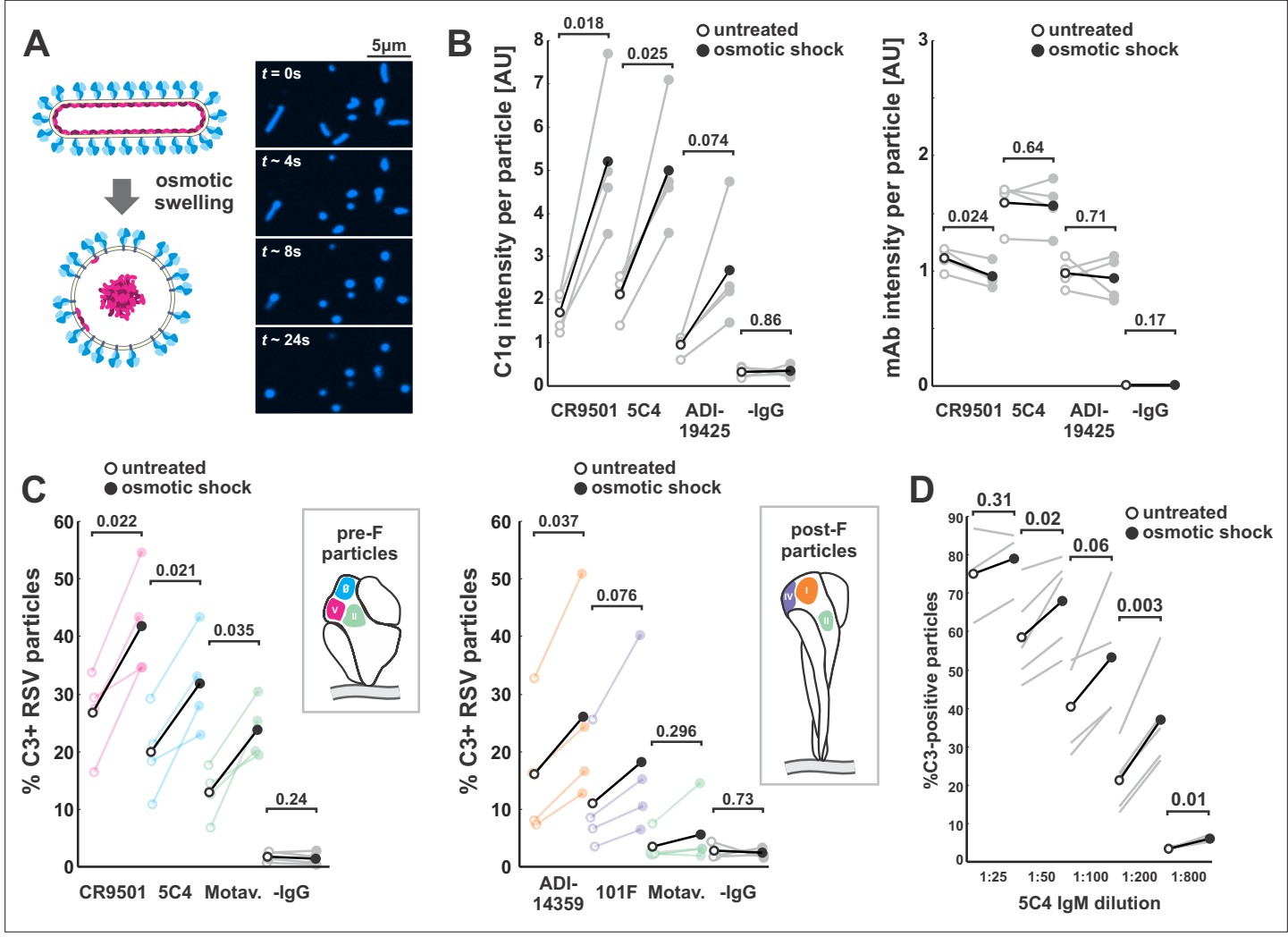

**Figure 6.** Membrane detachment from the viral matrix increases complement activation. (**A**) Process for detaching the RSV matrix from the viral membrane using osmotic swelling. Fluorescence images show details of the transformation, with t = 0 s approximately corresponding to the addition of low osmolarity buffer. (**B**) *Left*: C1q binding per RSV particle, with (filled circles) and without (open circles) osmotic swelling. Black markers indicate averages across four biological replicates. Individual replicates are shown in gray, with connecting lines indicating paired replicates. C1q intensity is measured using a C1qA-specific antibody conjugated with Alexa Fluor 488. *Right*: Average mAb intensity per particle for the same experiments plotted to the left. p-Values determined using a paired-sample t-test. (**C**) Percent C3-positive RSV particles with (filled circles) and without (open circles) osmotic swelling. Plots to the left are for predominantly pre-F particles; plots to the right are for predominantly post-F particles. Black markers indicate averages of four biological replicates (individual replicates are color-coded according to the mAb's antigenic site). p-Values determined using a paired-sample t-test. (**D**) Percent C3-positive RSV particles with (filled circles) and without (open circles) osmotic swelling across different dilutions of 5C4 IgM. Black lines indicate averages across individual biological replicates (shown in gray). p-Values determined using a paired-sample t-test. See also *Figure 6—source data 1*.

The online version of this article includes the following figure supplement(s) for figure 6:

**Source data 1.** Matlab code and data for *Figure 6*, *Figure 6—figure supplement 1*, and *Figure 6—figure supplement 2*.

**Figure supplement 1.** Osmotic swelling detaches the RSV matrix from the viral membrane with no loss of infectivity.

**Figure supplement 2.** C3 deposition on RSV particles increases with decreasing particle curvature.

**Figure supplement 3.** A conceptual model for the changing surface of RSV.

we compared C3:F ratios between osmotically swollen (spherical) particles and non-treated (filamentous) particles from the same initial population of RSV. While C3:F ratios are similar for smaller viruses regardless of swelling, C3:F is increased by up to 9-fold upon swelling in larger viruses, indicating a preference for lower curvature in IgG1-driven deposition of complement proteins (*Figure 6—figure supplement 2C*). This effect is unlikely to arise due to changes in the abundance or density of F in the

viral membrane, both of which will remain constant following swelling. Similarly, it does not appear to be purely related to size, as larger viral filaments show similar C3:F ratios as smaller viral filaments. Thus, filamentous RSV particles are poised to evade activation of the classical pathway when they emerge from cells, but become substantially more susceptible as they age and their shape begins to change (*Figure 6—figure supplement 3*).

## High surface curvature decreases the effective density of membrane-distal epitopes

To better understand the origin of this curvature dependence, we developed a course-grained model based on idealized geometries of RSV filaments, F, and hexameric IgG (*Figure 7A*). Although this model is highly reductionist, it allows us to examine the effects of variables such as the distance between neighboring epitopes which we are unable to control experimentally. Rather than modeling antibody binding and assembly into hexamers explicitly as a two-step process, we instead started from pre-formed IgG hexamers and asked how readily these hexamers could be accommodated by a particular surface: that is, we sought to determine the percentage of randomly sampled hexamer configurations that allow six Fabs to simultaneously engage with antigen on a surface (*Figure 7B*). We reasoned that surfaces on which a higher percentage of hexamer configurations fit will be more likely to support activation of the classical pathway. Consistent with our experimental results, these simulations predict that decreasing curvature allows a higher proportion of hexamer configurations to be accommodated. The ability to accommodate hexamers is also extremely sensitive to the spacing between epitopes in the membrane, a trend that we observe for multiple organization of the F lattice (*Figure 7C*).

The observation that shorter distances between neighboring epitopes helps accommodate hexameric IgG suggests an intuitive explanation for how surface curvature may influence activation of the classical pathway. As surface curvature increases, the distance between epitopes located a fixed height above the membrane also increases due to the splaying of surface proteins (*Figure 7D*, left). This effect should be stronger for epitopes further above the viral membrane. Consistent with this prediction, our simulations show that epitopes in the plane of the membrane are less sensitive to curvature than those 12 nm above the membrane, approximating the height of RSV F (*Figure 7D*, right). Accordingly, surface curvature may influence activation of the classical pathway indirectly, by changing the spacing between membrane-distal epitopes. The generality of this proposed mechanism suggests that it may influence activation of the classical pathway by other enveloped viruses as well.

## Discussion

Activation of complement during RSV infection has been linked to both protective and pathogenic effects, but the mechanisms that drive complement activation by RSV remain unclear. We find that a number of factors contribute to activation of the classical pathway by shed RSV particles, including characteristics of the targeting antibody, the packaging of complement defense proteins into the viral membrane, and biophysical characteristics of RSV particles. Among these factors, the dominant contribution comes from the activating antibody. We find an antigenic hierarchy in the ability of different mAbs to drive complement deposition: although all of the IgG1 antibodies we have tested contain the same Fc and are capable of binding to pre- or post-F at similar antibody:F ratios, only a subset lead to appreciable deposition of C3. A common feature of these more potent antibodies is that they are all predicted to project their Fc above the surrounding canopy of F (>15 nm above the viral membrane), leading to more efficient binding by C1 (*Figures 1 and 2*). The typical distance between adjacent F trimers in the RSV membrane appears to be ~10–20 nm (*Ke et al., 2018a*), a value that is roughly consistent with other filamentous viruses (*Wasilewski et al., 2012*; *Ke et al., 2018b*; *Beniac and Booth, 2017*). Since this spacing is too small to accommodate an activating IgG or IgM platform between F trimers and since F appears to be largely stationary within the membrane of viral filaments (*Figure 6—figure supplement 1C and D*), it may be critical to position antibody Fcs in this way so as to avoid steric clashes with adjacent F trimers. A consequence of this scenario is that activation of C1 would occur at comparatively large distances from the membrane. While this could potentially limit the proper assembly of a membrane attack complex where proximity to the membrane may be

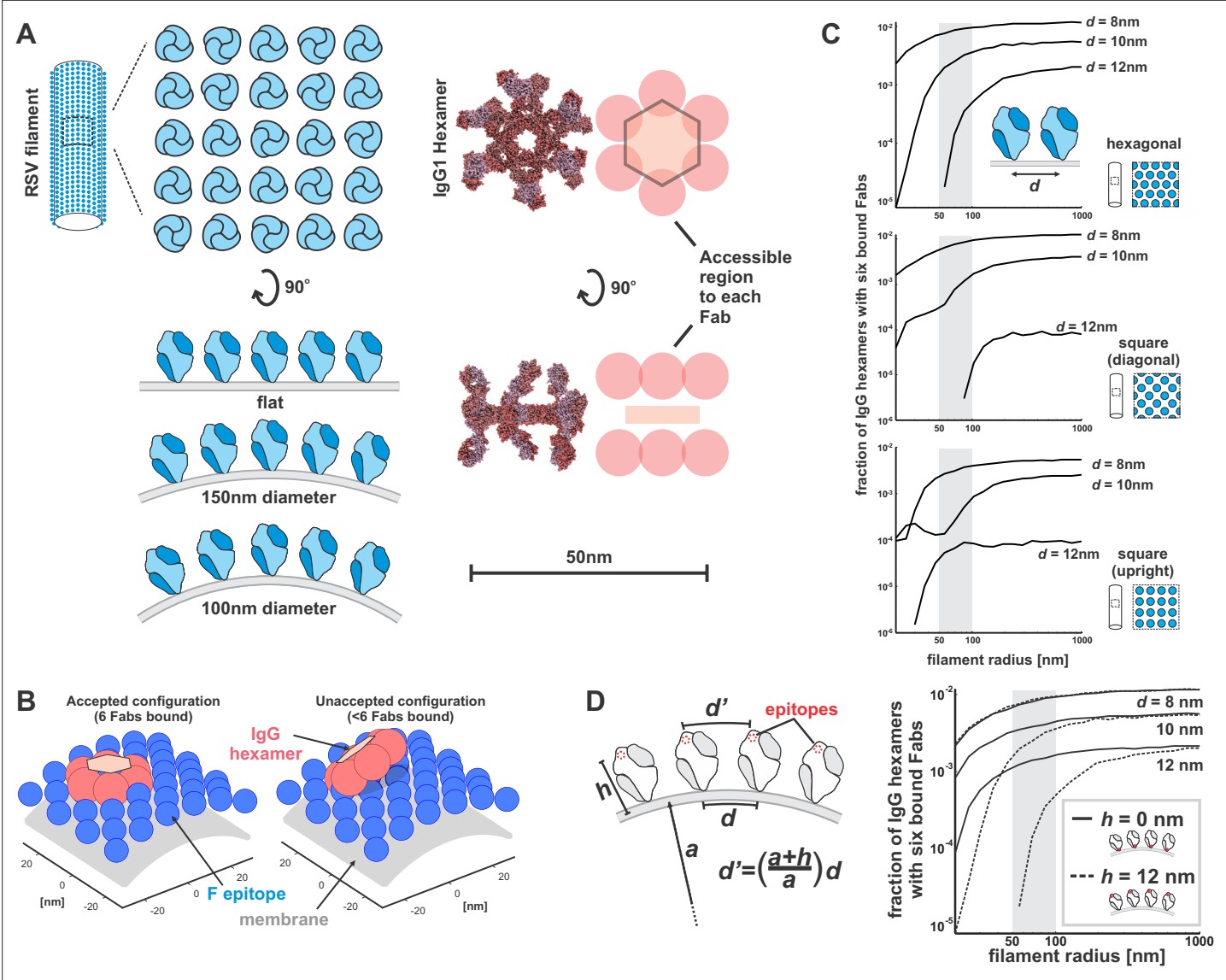

**Figure 7.** RSV curvature constrains docking of IgG hexamers. (**A**) Schematic of RSV F on virion surfaces with varying curvature (*left*) and a structure (PDB ID 1HZH) and simplified model of an IgG1 hexamer (*right*). The curved membrane, F, and hexameric IgG are drawn approximately to scale relative to the 50 nm scale bar. (**B**) Snapshots of simulations showing an accepted hexamer configuration (*left*) and an unaccepted configuration (*right*) on a surface with 50 nm radius. For clarity, only Fabs facing the surface are shown. (**C**) Model predictions for the fraction of IgG hexamer configurations that are compatible with surfaces of different curvature (plotted as filament radius on the horizontal axis). Simulations consider different lattice organization (hexagonal, square-diagonal, and square-upright), as well as different epitope spacing (d), as measured in the plane of the membrane. The shaded region in the plots corresponds to the expected range of curvatures observed for filamentous RSV particles. (**D**) *Left*: A proposed model connecting membrane curvature to effective epitope density. Higher curvatures (corresponding to smaller values of the radius, a) result in greater splay of F trimers in the membrane, increasing the distance between membrane-distal epitopes (d'). *Right*: Simulations comparing IgG hexamer docking to epitopes above the membrane (h = 12 nm, as in C; dashed lines), or within the plane of the membrane (h = 0 nm; solid lines). This model does not account for possible steric hindrance that could arise in the case of membrane-proximal epitopes. See also *Figure 7—source data 1* and *Figure 7—source code 1*.

The online version of this article includes the following figure supplement(s) for figure 7:

**Source code 1.** Matlab code for running simulations for *Figure 7*.

**Source data 1.** Matlab code and data for *Figure 7*.

beneficial (*Doorduijn et al., 2020*), it may be inconsequential to other aspects of activation including the production of C3a, which has been implicated in RSV disease severity (*Bera et al., 2011*).

Additional factors not considered here may also contribute to complement responses during in vivo infection or vaccination. One example is the effects of polyclonal serum. The monoclonal antibodies used in our experiments are present at concentrations similar to that of dominant clonotypes in post-vaccination serum (~10–20 μg/ml) (*Lavinder et al., 2014*) – well above the effective $K_D$s of these mAbs. At these relatively high concentrations, the amount of bound antibody per virus is limited by the available epitopes on the virus rather than by the binding affinity of the antibody. In polyclonal serum, antibodies that bind to non-overlapping epitopes could further increase the total density of antibody bound to viruses. This could facilitate the formation of Fc hexamers comprised of different antibody clonotypes. Delineating how complement activation driven by polyclonal responses compares to the monoclonal conditions studied here will require additional work. A second factor not captured by our experiments is antibody-induced particle aggregation. We specifically performed experiments on immobilized RSV particles to enable us to determine characteristics of individual viruses that do or do not drive complement deposition. Performing similar experiments in solution, where larger and more heterogeneous immune complexes could form, would further improve understanding of factors that drive the complement response to RSV. Finally, we note that our experiments are restricted to surface curvatures within the range biologically accessible to RSV. Experiments with engineered systems – including protein nanoparticle vaccines with much higher curvature than native RSV particles (*Marcandalli et al., 2019*) – may help to determine if some epitopes exhibit greater ability to drive complement deposition in this regime.

We developed geometry-based simulations to better understand how curvature may influence the binding of C1 by supporting hexameric configurations of IgG. This model corroborates the curvature sensitivity of IgG hexamer formation and suggests a possible intuitive explanation for our findings: high curvature effectively increases the distance between epitopes that sit high above the viral membrane, decreasing the likelihood of hexamer formation. This model suggests that the further the epitope is above the viral membrane, the greater the effect that decreasing curvature will have. Although this would predict that epitopes closest to the membrane may be best suited to activate the classical pathway, we find that this is not the case. Antibodies that bind to more membrane-proximal epitopes on pre-F (e.g. 101 F or ADI-19425) are overall very inefficient at driving complement deposition (*Figure 1*). We attribute this to steric obstruction of the Fc regions of these antibodies, which prevents them from forming activating hexamers (*Figure 2*). Thus, there may be an inherent tradeoff between overcoming steric obstruction (by binding to epitopes distal to the membrane) and sensitivity to surface curvature in activation of the classical pathway.

Perhaps the most unexpected result presented here is the observation that changes in virus shape can both increase the deposition of complement proteins (*Figures 4–6*) and lead to more rapid accumulation of post-F in the viral membrane (*Figure 6—figure supplement 1B*). Collectively, these substantial changes in the molecular surface of RSV particles may influence their interactions with immune cells. Given the importance of C3 opsonization in antigen presentation and B cell activation (*Dempsey et al., 1996*), we speculate that our results may contribute to the focusing of adaptive responses to post-F antigens, which we find are more abundant on particles that accumulate C3 (*Figure 5*). Consistent with this model, previous work has established that infants mount an adaptive immune response to RSV focused largely on prefusion F (specifically, via site III), and that this response changes over time to include post-F epitopes as the infants age (*Goodwin et al., 2018*). While the connection between these prior observations and the results presented here is speculative, investigating whether biases in complement activation contribute to this observed shift in immunodominance toward post-F epitopes may be warranted.

Antibody responses specific to postfusion F have been linked to increased disease severity associated with enhanced complement activation (*Polack et al., 2002*; *Acosta et al., 2015*). Post-F antibodies are poorly neutralizing and generally ineffective at controlling infection. This was observed in trials for a formalin-inactivated RSV vaccine, where formalin treatment converted pre-F to post-F and resulted in enhanced disease among vaccine recipients upon subsequent infection (*Kim et al., 1969*; *Killikelly et al., 2016*). Our results may help to further understand the significance of post-F antibodies that arise as a consequence of vaccination or natural infection. Specifically, we observe a high correlation between globular morphology and enrichment of post-F (*Figure 5*). The high intrinsic capacity

of globular particles to activate complement may therefore contribute to complement dysregulation caused by post-F-specific antibodies. Although more work is needed to understand the prevalence of globular particles during in vivo infection, our observations that these particles accumulate over time through the conversion of viral filaments – even under normal cell culture conditions – suggest that their presence in vivo is feasible, where the physical and chemical environment would be considerably harsher and more complex. Globular RSV particles would also likely be prevalent in vaccine preparations such as those used in the failed formalin-inactivated RSV vaccine trial, potentially contributing to the responses they induce. Overall, we propose that particle morphology may be relevant to immune responses to RSV and other viruses, and thus an important consideration in the development of vaccines using inactivated or attenuated virus as antigens.

The methodology presented here – combining site-specific labeling and fluorescence characterization to dissect mechanisms of complement activation by RSV – provides a model that could be generalized to other pathogens. Although investigating complement activation in vitro limits our ability to assess direct immunological consequences of activation, it provides the ability to isolate the contributions that different antibodies and distinct particle subsets make toward complement activation. This may be particularly useful for viruses that vary widely in the biophysical characteristics of released progeny, including RSV, HMPV, influenza, and Ebola. Extending this methodology to investigate other viruses and their interactions with immune receptors could help determine if particle heterogeneity contributes to disparate outcomes during immune signaling in other contexts as well.

# Materials and methods

**Key resources table**

| Reagent type (species) or resource | Designation | Source or reference | Identifiers | Additional information |
|---|---|---|---|---|
| Cell line (*Homo sapiens*) | HEK 293T | ATCC | ATCC Cat# CRL-3216, RRID:CVCL_0063 | |
| Cell line (*Homo sapiens*) | A549 | ATCC | ATCC Cat# CCL-185, RRID:CVCL_0023 | |
| Cell line (*Melanochromis auratus*) | BHK-21 | ATCC | ATCC Cat# CCL-10, RRID:CVCL_1915 | |
| Strain, strain background (*Escherichia coli*) | SW102 | NCI Preclinical Repository | | Used for BAC recombineering |
| Recombinant DNA reagent | RSV BAC | BEI Resources | NR-36460 | This BAC has been further modified for this work as described in Materials and methods |
| Recombinant DNA reagent | pA2-Lopt | BEI Resources | NR-36461 | |
| Recombinant DNA reagent | pA2-Nopt | BEI Resources | NR-36462 | |
| Recombinant DNA reagent | pA2-Popt | BEI Resources | NR-36463 | |
| Recombinant DNA reagent | pA2-M2-1opt | BEI Resources | NR-36464 | |
| Recombinant DNA reagent | pCAGGS-T7 RNAP | This work | | Plasmid encoding T7 RNAP |
| Recombinant DNA reagent | pmAb-ADI-14353 HC | This work | | Plasmid for expressing recombinant antibody |
| Recombinant DNA reagent | pmAb-ADI-14353 LC | This work | | Plasmid for expressing recombinant antibody |
| Recombinant DNA reagent | pmAb-ADI-14359 HC | This work | | Plasmid for expressing recombinant antibody |

*Continued on next page*

*Continued*

| Reagent type (species) or resource | Designation | Source or reference | Identifiers | Additional information |
|---|---|---|---|---|
| Recombinant DNA reagent | pmAb-ADI-14359 HC Fab | This work | | Plasmid for expressing recombinant antibody |
| Recombinant DNA reagent | pmAb-ADI-14359 LC | This work | | Plasmid for expressing recombinant antibody |
| Recombinant DNA reagent | pmAb-ADI-19425 HC | This work | | Plasmid for expressing recombinant antibody |
| Recombinant DNA reagent | pmAb-ADI-19425 LC | This work | | Plasmid for expressing recombinant antibody |
| Recombinant DNA reagent | pmAb-5C4 HC (IgG1) | This work | | Plasmid for expressing recombinant antibody |
| Recombinant DNA reagent | pmAb-5C4 HC (IgM) | This work | | Plasmid for expressing recombinant antibody |
| Recombinant DNA reagent | pmAb-5C4 LC | This work | | Plasmid for expressing recombinant antibody |
| Recombinant DNA reagent | pmAb-CR9501 HC | This work | | Plasmid for expressing recombinant antibody |
| Recombinant DNA reagent | pmAb-CR9501 LC | This work | | Plasmid for expressing recombinant antibody |
| Recombinant DNA reagent | pmAb-Motavizumab HC | This work | | Plasmid for expressing recombinant antibody |
| Recombinant DNA reagent | pmAb-Motavizumab LC | This work | | Plasmid for expressing recombinant antibody |
| Recombinant DNA reagent | pmAb-101F HC | This work | | Plasmid for expressing recombinant antibody |
| Recombinant DNA reagent | pmAb-101F LC | This work | | Plasmid for expressing recombinant antibody |
| Recombinant DNA reagent | pmAb-D25 HC Fab | This work | | Plasmid for expressing recombinant antibody |
| Recombinant DNA reagent | pmAb-D25 LC | This work | | Plasmid for expressing recombinant antibody |
| Recombinant DNA reagent | pmAb-3D3 HC | This work | | Plasmid for expressing recombinant antibody |
| Recombinant DNA reagent | pmAb-3D3 LC | This work | | Plasmid for expressing recombinant antibody |
| Recombinant DNA reagent | pmAb-J chain | This work | | Plasmid for expressing recombinant antibody |
| Recombinant DNA reagent | Lenti CRISPRv2 | Addgene | | Ref. [70] |
| Biological sample (*Homo sapiens*) | IgG/IgM-depleted normal human serum | Pelfreeze | 34,014 | |
| Biological sample (*Homo sapiens*) | C3 | Complement Technology, Inc | A113 | |
| Biological sample (*Homo sapiens*) | C4 | Complement Technology, Inc | A105 | |
| Biological sample (*Homo sapiens*) | C1 | Complement Technology, Inc | A098 | |
| Antibody | Mouse anti-CD46 monoclonal (TRA-2–10) | BioLegend | BioLegend Cat# 352404, RRID:AB_10900243 | IF (1:500) |

*Continued on next page*

*Continued*

| Reagent type (species) or resource | Designation | Source or reference | Identifiers | Additional information |
|---|---|---|---|---|
| Antibody | Mouse anti-CD55 monoclonal (JS11) | BioLegend | BioLegend Cat# 311301, RRID:AB_314858 | IF (1:500) |
| Antibody | Mouse anti-C1q monoclonal (1A4) | Santa Cruz Biotech. | Santa Cruz Biotechnology Cat# sc-53544, RRID:AB_1119798 | IF (1:500) |
| Peptide, recombinant protein | Streptavidin | Thermo Fisher Scientific | 434,302 | |
| Peptide, recombinant protein | CLPMTGG peptide | Genscript | | Peptide for SrtA-based labeling |
| Peptide, recombinant protein | Sortase A | This work | | Recombinant SrtA for labeling |
| Peptide, recombinant protein | Sfp synthase | This work | | Recombinant Sfp for labeling |
| Chemical compound, drug | Sulfo-Cy5 maleimide | Lumiprobe | Cat# 13,380 | |
| Chemical compound, drug | AF-488 NHS Ester | Lumiprobe | Cat# 11,820 | |
| Chemical compound, drug | Alexa Fluor 488 C5 maleimide | Thermo Fisher Scientific | Cat# A10254 | |
| Chemical compound, drug | Alexa Fluor 555 C2 maleimide | Thermo Fisher Scientific | Cat# A20346 | |
| Chemical compound, drug | NH2-PEG-Biotin | Rapp Polymere | Cat# 133000-25-20 | |
| Chemical compound, drug | CH$_3$O-PEG-NH2 | Rapp Polymere | Cat# 122000–2 | |
| Chemical compound, drug | Coenzyme A trilithium salt | Sigma-Aldrich | Cat# C3019 | |
| Software, algorithm | Matlab R2020a | Mathworks | | Used for data analysis, plotting, and simulations |
| Software, algorithm | Nikon Elements | Nikon | | Used for image acquisition |
| Software, algorithm | PyMOL | Schrödinger, Inc | | Used for structure alignment |

## Creating recombinant RSV for site-specific labeling

We introduced modifications into the RSV genome following the approach of *Hotard et al., 2012* . Briefly, we electroporated a BAC containing the antigenomic cDNA of a chimeric RSV strain A2 with the F protein from Line 19 (obtained through BEI Resources) into the *Escherichia coli* strain SW102 (*Warming et al., 2005*). To avoid virus attenuation and fluorescence spectral overlap caused by the mKate2 reporter in front of NS1 in the initial construct, we replaced it with an mTagBFP2 reporter, expressed from an IRES following NS1. From this modified BAC, we proceeded to use a galK cassette with a 5' homology arm targeting the C-terminal region of G and a 3' homology arm targeting the N-terminal region of F to insert tags on G and F simultaneously following an initial round of selection on galactose plates. We verified successfully modified BACs by sequencing a PCR-amplified region around the modified site, and we purified the BACs from 250 ml cultures using a Nucleobond BAC100 kit. The genomic sequence for the final virus is given in *Supplementary file 1*.

To rescue recombinant viruses, we transfected 6-well plates of BHK-21 cells with BAC (0.8 µg), helper plasmids (codon-optimized L, N, P, and M2-1 at 0.2, 0.4, 0.4, and 0.4 µg, respectively), and a vector containing the T7 RNA polymerase (0.2 µg) using Lipofectamine 2000. Helper plasmids were obtained from BEI Resources. Following transfection, cells cultured in virus growth media (OptiMEM with 2 % FBS and antibiotic-antimycotic) were passaged every 2–3 days at a 1:3 ratio and monitored for signs of infection. To collect virus stocks, we removed media from T75 flasks of infected

cells and replaced it with 2 ml of PBS supplemented with 2 mM EDTA. Cells that detached from the flask were collected, flash-frozen in liquid nitrogen, and pelleted to remove cell debris following a rapid thaw at 37 °C. The virus-containing supernatant was then aliquoted and stored at –80 °C until needed for experiments. Detailed characterization of the replication of this recombinant strain will be published elsewhere. Viral titers used for MOI calculations were determined by quantifying mTagBFP2-expressing A549 cells infected in 96-well plates as a function of the input volume of virus.

### Preparing viruses

We used viral stocks snap-frozen and stored at –80 °C to infect ~90 % confluent A549 cells in eight-chambered coverglass or 96-well plates at MOI ~ 1. To enhance the efficiency of infection, virus diluted into virus growth media (final volume of 100 µl) was centrifuged onto cells at 1200×g for 10 min and returned to the incubator for an additional 50 min before washing off the virus-containing media and replacing with fresh virus growth media. Samples used for independent biological replicates were conducted using viruses collected from independent infections.

Enzymes and probes for site-specific labeling were generated as previously described (*Vahey and Fletcher, 2019a*; *Vahey and Fletcher, 2019b*). At 48–60 hr post infection, F on the surface of infected cells was labeled in situ using 50 µM SrtA and 100 µM fluorescent CLPMTGG substrate. With the exception of photobleaching experiments (where sulfo-Cy3 was used), sulfo-Cy5 maleimide (Lumiprobe, 13380) was conjugated to the N-terminal cysteine of the SrtA peptide to label F. Labeling reactions were prepared in virus growth media supplemented with 5 mM $CaCl_2$. For experiments with labeled G, the labeling reaction also included 5 µM Sfp, 10 µM CoA-probe, and 5 mM $MgCl_2$. After labeling cell surface viral proteins for 2 hr at room temperature, the labeling reaction was washed four times with fresh media and the cells were returned to the incubator for an additional 2 hr to allow labeled viruses to detach. Viruses prepared in this way exhibited predominantly filamentous morphology (~70%) and fewer than 5 % contained detectable post-F.

### Coverslip functionalization and virus immobilization

Coverslips for virus immobilization and imaging were prepared following a PEGylation protocol modified from *Piehler et al., 2000*. Briefly, glass coverslips were cleaned with sonication using a 50 % ethanol/50 % 3 M NaOH solution for 30 min, followed by two rinses in 1 l beakers of milliQ water. Coverslips were then cleaned using piranha solution (60 % sulfuric acid, 40 % $H_2O_2$) and sonication for 45 min, rinsed, dried, and functionalized with (3-glycidyloxypropyl)trimethoxysilane (GOPTS) for 1 hr at 75 °C. Excess GOPTS was rinsed from coverslips using anhydrous acetone, and a mixture of biotin-PEG-amine/methoxy-PEG-amine (Rapp Polymere) was prepared in anhydrous acetone at a ratio of 10 mol% biotinylated PEG. The PEG solution was coupled to coverslips overnight at 75 °C, rinsed twice in 1 l beakers of milliQ water, and stored in milliQ water at 4 °C until use.

For virus immobilization, PEGylated coverslips were rinsed in ethanol, dried, and sealed with custom chambers made of polydimethylsiloxane with wells shaped using a 4 mm biopsy punch. Wells were filled with PBS and incubated successively with streptavidin (5 µg/ml in PBS) and anti-G antibody 3D3 (*Collarini et al., 2009*) with a biotin site-specifically conjugated to the C-terminus of the heavy chain (see *Antibody cloning, expression, purification, and labeling*). After washing wells 10 times with PBS to remove excess antibody, coverslips were stored at 4 °C in a humidified enclosure for <2 days, until ready for use.

### Antibody cloning, expression, purification, and labeling

Antibody sequences used in this work are listed in *Supplementary file 1*. VH and VL sequences were cloned into a human IgG1 backbone with a C-terminal ybbR tag using Gibson assembly. Verified clones were used to transfect T75 flasks of HEK293s at ~85 % confluency. At ~12 hr post transfection, cells were washed twice with PBS to remove any residual IgG from the serum-containing culture media and grown for an addition 6 days in serum-free OptiMEM. Media containing secreted mAbs was collected and centrifuged at 1000×g to remove detached cells before purification with Protein G resin. Fab fragments (ADI-14359, D25) and 5C4 IgM were expressed and purified analogously, with the exception that a C-terminal His(6)-tag on the heavy chains were used for affinity purification by Ni-NTA agarose in place of Protein G resin.

Eluted antibodies were quantified, diluted into a new buffer for enzymatic labeling (150 mM NaCl, 25 mM HEPES, 5 mM MgCl$_2$), and concentrated using centrifugal concentrators (VIVAspin 100 K). Antibodies concentrated to ~1 mg/ml were then labeled overnight on ice using Sfp synthase and CoA-conjugated dyes, prepared as previously described (*Vahey and Fletcher, 2019a*). Following removal of excess dye using PD-10 desalting columns, labeling efficiencies were determined spectro-photometrically to be >90% for all antibodies based on the number of heavy chains. For virus immo-bilization, we cloned and purified the anti-G antibody 3D3 (*Fedechkin et al., 2018*; *Collarini et al., 2009*) using the same protocol, but substituting CoA-biotin for the fluorescent dyes.

## C3/C4 deposition assay

C3 deposition assays were performed using IgG/IgM-depleted normal human serum (NHS; Pelfreeze 34014). Fluorescent C3 and C4 were produced by labeling purified C3 or C4 (Complement Technol-ogies, A113 and A105) using AF-488 dye functionalized with *N*-hydroxysuccinimide ester (Lumiprobe 11820). Labeling reactions were calibrated to prevent over-labeling of proteins and resulted in ~0.6–1.0 dye molecules/protein, as determined via spectrophotometry. Complement reactions to monitor C3 deposition were prepared using complement buffer (150 mM NaCl, 25 mM HEPES, 0.5 mM MgCl$_2$, 0.15 mM CaCl$_2$) supplemented with 10 mg/ml BSA, 5 % IgG/IgM-depleted NHS, 50 µg/ml 488-C3, and 10–20 µg/ml fluorescent mAb (to assure rapid saturation of binding). Assuming a C3 concentra-tion of 1 mg/ml in NHS, ~30–50% of C3 in the experiment will carry a fluorophore. Prior to starting the reaction, all samples were washed thoroughly with complement buffer. Complement reactions were prepared on ice and added to virus samples before incubation at 37 °C/5 % CO$_2$/100 % humidity for 1 hr. Following incubation, samples were washed three times with PBS to terminate the reaction and imaged immediately. A similar procedure was followed for timelapse experiments using C4, except NHS was used at a final concentration of 2.5 % and 488-labeled C4 was supplemented at 10 µg/ml. To synchronize recording with the start of the reaction, samples containing immobilized RSV were mounted in an incubated enclosure on the microscope prior to adding the complement components.

## C1-binding assay

C1-binding assays were performed using purified C1 (Complement Technologies A098) in the absence of other serum proteins and the presence of defined mAbs. C1 was diluted into complement buffer to a final concentration of 10 µg/ml. The reaction also contained 10 mg/ml BSA and saturating concen-trations of defined mAbs (10–20 µg/ml, as in C3 deposition assays). This mixture was incubated with enzymatically labeled RSV particles for 90 min at 4 °C, washed three times in complement buffer, and labeled with an Alexa Fluor 488-conjugated anti-C1qA antibody (1A4; Santa Cruz sc-53544 AF488) at a 1:500 dilution in complement buffer for 30 min at room temperature before imaging.

## Fluorescence microscopy and image analysis

Following incubation with antibodies and complement components, we imaged opsonized RSV parti-cles using a 60×, 1.40 NA objective on a Nikon T2i microscope body equipped with a Yokogawa CSU-X spinning disk and ORCA-Flash4.0 V3 camera. For each sample condition per biological repli-cate, we collected images of ~15 randomized fields of view, each containing ~500–1000 RSV particles. The resulting image datasets were analyzed and plotted using custom Matlab scripts. Images were segmented using the F channel to identify pixels associated with each virus in the image. Following background subtraction, fluorescence intensities were integrated across all channels to obtain an inte-grated intensity for F, mAb, and C3/C4/C1 for each segmented particle. Integrated intensities were then plotted directly to determine population distributions (e.g. *Figure 1C*), or simplified further by determining the percentage of positive particles (e.g. *Figure 1D*).

## Determining linearity of fluorescence measurements

To verify that our data collection falls within a linear regime, we used a D25 Fab (*McLellan et al., 2013b*), labeled in a precise 1:1 (Fab:dye) ratio via a C-terminal ybbR tag attached to the heavy chain fragment. We selected the D25 Fab for these experiments for several reasons: its ability to bind F protomers 1:1 at high concentrations; the accessibility of its epitope even on a crowded virion surface; and its exceptional affinity and slow dissociation. These characteristics allow us to image RSV parti-cles with saturating amounts of bound D25 while still preserving low background fluorescence. We

labeled D25 Fabs in two different colors using AF488 and sulfo-Cy5 dyes (excited at 488 and 640 nm, respectively). Combining these labeled Fabs in defined ratios allows us to compare intensity values to what we would expect to see based on the input ratio of labeled Fab. Measurements obtained using microscope settings matching those used throughout this work indicate that fluorescence measurements are linearly proportional to RSV-bound fluorophores across at least an ~250- fold range for absolute measurements, and across an ~60,000 -fold range for relative measurements (*Figure 1— figure supplement 2*).

### Determining enzymatic labeling efficiencies of RSV F

To estimate the fraction of F that is fluorescently labeled by SrtA in our experiments, we used a strategy analogous to our assessments of linearity. First, we bound RSV particles with equal amounts of D25(AF488) and D25(Cy5) Fab. Imaging these particles allows us to determine the relative intensities of these two dyes under our imaging settings. We then use this calibration to compare the relative number of dye molecules present on RSV dually labeled with sulfo-Cy5 via SrtA and AF488 via the D25 Fab. This analysis yields an estimated labeling efficiency of ~90 % across two biological replicates (*Figure 1—figure supplement 3A and B*). To corroborate this analysis, we directly compared intensities of viruses labeled with sulfo-Cy5 via SrtA to viruses labeled with sulfo-Cy5 via D25 Fab (*Figure 1—figure supplement 3C*). This approach yields similar results, with the SrtA sample giving a median intensity 1.04 and 1.18 times brighter than the D25 Fab across two biological replicates, supporting the conclusion that SrtA labeling of RSV-F on the surface of virions is nearly complete. In these comparisons, values that are modestly greater than 1 could arise due to the specificity of D25 for pre-F, vs. the ability of SrtA to label both pre- and post-F.

### Virus photobleaching

Viruses labeled with Sulfo-Cy3 CLPMTGG via SrtA were immobilized on coverslips using antibodies against G (3D3) and imaged on an Olympus FluoView FV1200 laser scanning confocal microscope using a 60×, 1.35 NA objective. A circular region ~1 μm in diameter that overlapped with a portion of the virus particle was selected and bleached using a 561 nm laser, and fluorescence recovery was monitored by imaging at 5 s intervals for 1 min, including one image pre-bleach and one image immediately post-bleach. To identify globular particles enriched in post-F, a 488-labeled Fab fragment with VH and VL domains from ADI-14359 was added as a marker. Use of a Fab fragment for these experiments prevented antibody-mediated crosslinking of F that could alter fluorescence recovery. Time series of bleaching and recovery were used to determine differences in F mobility. The percentage recovery was determined by generating an image mask from the difference between the first frame post-bleach and the last frame pre-bleach, to identify the bleached pixels. Intensities within the masked region were then integrated to quantify signal before bleaching, immediately after bleaching, and after a 20 s recovery.

### Infectivity comparisons

Comparisons of virus infectivity following various treatments (*Figure 1—figure supplement 1B*, *Figure 6—figure supplement 1A*) used expression of the mTagBFP2 reporter to quantify infected cells. For comparisons of infectivity with or without fluorophores conjugated to F, viruses were labeled at 60 hpi as described under *Preparing viruses*. Control samples were incubated at room temperature in parallel with labeled samples, and washed in the same way, to assure that collected viruses in all cases were shed exclusively over a 2 hr period; 5 μl of these samples were used to infect confluent A549 cells in 96-well plates, as described in *Preparing viruses*. Infection was quantified by counting BFP-positive cells across ~10 fields of view at 10 × magnification at 12 hpi. Virus quantified in this way reflects only the particles shed from cells in a 2 hr period, and does not reflect the potentially substantial fraction of virus that remain cell-associated, or that were removed during wash steps.

For comparisons of untreated and osmotically swollen viruses, 96-well plates containing infected A549 cells at 60 hpi were washed with fresh media to remove older virus and returned to the incubator for 2 hr to allow new virus to shed. Collected samples were then split into experimental and control groups. For control groups, 5 μl of shed virus was diluted into 95 μl of 1× MEM with sodium bicarbonate and 7.5 mM HEPES. For experimental groups, 5 μl of shed virus was diluted into 75 μl of 10 mM HEPES (the low osmolarity buffer used for osmotic swelling), incubated for ~1 min, and

added to 10 µl of 10 × sodium bicarbonate and 10 µl of 10 × MEM, so that the final composition of control and experimental groups is matching. Samples were then used to infect confluent A549 cells in 96-well plates seeded the previous day. Infection was quantified by counting BFP-positive cells across ~10 fields of view at 10 × magnification at 12 hpi.

## Creating polyclonal A549 CD55 and CD46 KO lines

A549 KO cells were generated through transduction with lentivirus generated from the lentiCRISPR v2 packaging plasmid (*Sanjana et al., 2014*). Three sgRNA sequences were selected using CRISPR KO and the design rules described by *Doench et al., 2016*. These were tested in small-scale via transient transfection in HEK293s and the sgRNAs that yielded the highest efficiency (determined via immunofluorescence) were selected for lentivirus preparation and infection into A549s. The spacer sequences used for CD55 and CD46 sgRNAs are 5'-GCACCACCACAAATTGACAA-3' (for CD55) and 5'-GTTTGTGATCGGAATCATACA-3' (for CD46; underline indicates a nucleotide added for efficient transcription initiation). Polyclonal KO cells were further enriched at the Washington University Flow Cytometry core, using a FACS Aria II to isolate cells negative for surface staining with fluorescent antibodies against CD46 (clone TRA-2–10) or CD55 (clone JS11).

## Modeling virus curvature

We modeled RSV particles in two simplified morphological states: filamentous particles – consisting of a cylindrical region of length L and radius $a_f$ and two hemispherical caps – and globular particles, which we approximate as spheres of radius $a_s$. During the transition from a filament to a sphere, the surface area of the virus remains constant; this is constrained by the number of lipids packaged during assembly and the inability of lipid membranes to withstand area strains above ~5% (*Needham and Nunn, 1990*). Conversely, the volume of the virus may change due to a flux of water into or out of the particle. The relationships in *Figure 6—figure supplement 2A* were obtained by applying a constant area constraint and equating the two surface areas (filament and sphere) and solving for the mean curvature in both cases.

## Modeling IgG hexamers on curved surfaces

To determine the ability of different surfaces to accommodate IgG1 hexamers, we built a simplified model from available structural information. In this model, antibody Fab arms are positioned adjacent to the vertices of a hexagon whose sides measure 9 nm in length (modeled from the IgG1 hexamer [*Saphire et al., 2001*], PDB ID 1HZH). We approximate the flexibility of the antibody Fab arms by modeling each as a spherical region with a radius of 5 nm; antigen that falls within this region is considered accessible for binding. To model RSV antigens, we assume that F trimers are positioned on lattices with different symmetry (upright square, diagonal square, or hexagonal), with each monomer separated by a distance 'd' ( = 8, 10, or 12 nm in different simulations) as measured in the plane of the membrane, and extending above the membrane to a height of 12 nm. The RSV surface itself is modeled as a cylinder with a radius 'a' that we vary across simulations. Assumptions of the size and spacing of RSV F are supported by observations from cryoelectron microscopy (*Ke et al., 2018a*) as well as structural characterization of the assembly of viruses with homologous matrix proteins (*Battisti et al., 2012*).

From these models of an IgG hexamer and the RSV surface, we place a hexamer in a random location in a 10-nm-thick region concentric to the viral surface. We assign a random orientation to the hexamer by sampling values for the three Euler angles from uniform random distributions. We then determine the nearest antigen to each of the six Fab fragments oriented toward the surface; if all six have an available antigen within a 5 nm radius, the sampled configuration is considered compatible with the surface. Otherwise, we consider the configuration to be incompatible with the viral surface at that location. Surfaces with different curvatures or different antigen distributions can then be compared based on the proportion of hexamer configurations that are compatible with that surface. The premise of this model is that the ability of a particular surface to accommodate pre-formed hexamers by allowing them to form a complete set of six contacts reflects (but is not equivalent to) the efficiency of hexamer formation on that surface by antibodies that bind independently.

## Cell lines

Cells used in this work were purchased as authenticated cell lines (STR profiling) from ATCC. All cells tested negative for mycoplasma (MycoAlert Mycoplasma Detection Kit, Lonza, LT07-418) throughout the duration of the project.

## Statistical methods and replicates

Replicates in this work are biological replicates, with each replicate representing virus raised from a single culture well, that we separately label and use for experiments. Statistical tests were performed using Matlab R2020a software. Source data and code including statistical tests are included in source data files accompanying each figure. This work does not use randomization or masking.

## Acknowledgements

The authors would like to acknowledge members of the Vahey Lab for feedback and technical consultation and Dr Ali Ellebedy for the antibody expression plasmid backbone. The following reagent was obtained through BEI Resources, NIAID, NIH: Bacterial Artificial Chromosome Plasmid pSynkRSV-I19F Containing Antigenomic cDNA from Respiratory Syncytial Virus (RSV) A2-Line19F, NR-36460. This work was supported by a Burroughs Wellcome Fund Career Awards at the Scientific Interfaces Grant and unrestricted funds from Washington University.

## Additional information

### Funding

| Funder | Grant reference number | Author |
|---|---|---|
| Burroughs Wellcome Fund | 1013923 | Michael D Vahey |

The funders had no role in study design, data collection and interpretation, or the decision to submit the work for publication.

### Author contributions

Jessica P Kuppan, Margaret D Mitrovich, Conceptualization, Formal analysis, Investigation, Methodology, Resources, Software, Visualization, Writing – original draft, Writing – review and editing; Michael D Vahey, Conceptualization, Data curation, Formal analysis, Funding acquisition, Investigation, Methodology, Project administration, Resources, Software, Supervision, Validation, Visualization, Writing – original draft, Writing – review and editing

### Author ORCIDs

Jessica P Kuppan http://orcid.org/0000-0002-5969-1876
Margaret D Mitrovich http://orcid.org/0000-0002-1302-2837
Michael D Vahey http://orcid.org/0000-0001-9453-4860

### Decision letter and Author response

Decision letter https://doi.org/10.7554/eLife.70575.sa1
Author response https://doi.org/10.7554/eLife.70575.sa2

## Additional files

### Supplementary files

• Supplementary file 1. This document contains the genomic sequence of the modified respiratory syncytial virus (RSV) strain used in this work as well as antibody sequences.

• Transparent reporting form

### Data availability

All data generated or analysed during this study are included in the manuscript and supporting files. Source data files have been provided for all Figures (1-7).

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
