## [Decision Letter]

**Acceptance summary:**

This work provides important insights into how the biophysical properties of viruses shape the development of immune responses, which will inspire future studies on this topic.

**Decision letter after peer review:**

Thank you for submitting your article "A morphological transformation in respiratory syncytial virus leads to enhanced complement activation" for consideration by *eLife*. Your article has been reviewed by 3 peer reviewers, including Leslie Goo as Reviewing Editor and Reviewer #1, and the evaluation has been overseen by Jos van der (Senior Editor). The following individual involved in review of your submission has agreed to reveal their identity: Kelly K Lee (Reviewer #2).

The reviewers have discussed their reviews with one another and are very favorable about the manuscript. However, there are a number of issues that need to be addressed, which we have outlined below to help you prepare a revised submission.

Essential revisions:

1) Please provide a rationale for monoclonal antibody concentrations used and discuss how these relate to physiological concentrations as well as the more complex polyclonal response that would be observed following vaccination and natural infection.

2) As quantitation of fluorescence intensities is crucial for this study, how did the authors ensure their detection was in a linear sensitivity range for the detectors for the various fluorescent channels? How is the "density of antigen" quantitated? What fraction of F or G is labeled?

3) Instead of "complement activation," which involves much more than individual binding of complement components, it is better to use more specific terminology that accurately describes was actually measured (i.e. fluorescently labeled complement component binding).

4) The authors conclude that larger, globular particles are more susceptible to C3 deposition. The mechanistic link between size/globular morphology and post-F

abundance, and the potential confounding effect of the latter on C3 deposition needs to be discussed in more detail.

5) Related to point 4), throughout the manuscript, the back and forth discussion on the relationship among virion morphology, post-F abundance, IgG concentration, and binding of complement components is convoluted. A streamlined description and/or illustration of the proposed working model(s) based on the data is needed.

6) As a major conclusion of the work is that curvature of the virus surface has an outsize impact on the ability of complement proteins to bind, some effort at modeling this effect based upon known structures is warranted.

7) The extent to which the results regarding virion morphology are biologically relevant needs to be better discussed, particularly with respect to design of vaccine candidates, including the failed formalin-inactivated RSV vaccine that was enriched in post-F.

*Reviewer #1 (Recommendations for the authors):*

To address minor weaknesses above, a schematic to illustrate the working model(s) would be helpful.

*Reviewer #2 (Recommendations for the authors):*

1) I appreciate the clarity of the figures!

2) Error bars in Figure 2 C, D. Figure 2C caption please clarify: "Connecting lines show data from paired biological replicates". Does the "paired replicates" here refer to the experiment performed two independent times from separate virus preparations? Or as in other experiments 3 biological replicates were performed? Or does the pair refer to the paired experiments such as +/-CD55? In Figures 4 and 5 this was done, but here it is not clear. Please include error bars as well.

3) Figure 5D the grey individual replicates are so faint as to be almost invisible.

4) Some typos here and there. E.g. Figure 1 caption "positive", line 340 "asses".

*Reviewer #3 (Recommendations for the authors):*

1. The authors may want to include mention of the formalin-inactivated RSV vaccine candidate that caused vaccine-enhanced disease in a clinical trial that resulted in the deaths of 2 infants. Killikelly and Graham (Sci Rep. 2016 Sep 29;6:34108) showed that the formalin treatment caused all of the prefusion F to convert to postfusion F, and although not investigated in that study, the virion purification and incubation with formalin likely converted all particles to the pleiomorphic/spherical form.

2. The schematic in Figure 1A is incorrectly drawn since the 3D3 antibody that captures the virion to the coverslips should be binding to G and not F.

---

## [Author Response]

Essential revisions:1) Please provide a rationale for monoclonal antibody concentrations used and discuss how these relate to physiological concentrations as well as the more complex polyclonal response that would be observed following vaccination and natural infection.

We agree with the reviewers that our experiments using monoclonal antibodies are a substantial simplification of physiological circumstances, where vaccination or infection would give rise to a complex polyclonal response. To better place our experiments in this physiological context, we have included additional discussion of this point in the manuscript (see Line 125 of *Results* and the second paragraph of *Discussion*). We summarize our revisions below.

Dominant antibody clonotypes in serum, measured post-vaccination, have been estimated at total concentrations >10ug/ml, with single dominant clonotypes capable of reaching ~20ug/ml^1^. Similar values have recently been reported for the antibody response to two-dose mRNA vaccines against SARS-CoV-2^2^. These concentrations are similar to the concentrations of monoclonal antibodies used in our experiments (10-20μg/ml), which we have selected to facilitate comparisons across antibodies that differ widely in their binding affinity.

It remains an open question how activation of the classical pathway by polyclonal sera and monoclonal antibodies may differ. In the context of virus neutralization, recent work has demonstrated that polyclonal serum can be extremely focused, with most of the potency coming from one or more antibodies targeting a single epitope^3^. A similar situation may hold for complement activation as well, with a few dominant clonotypes (not necessarily neutralizing ones) driving the majority of the response. If this is the case, then complement deposition driven by monoclonal antibodies may not differ dramatically from that driven by polyclonal serum. However, we speculate that polyclonal responses may still differ in some important ways. We found that activation of the classical pathway is not restricted to antibodies against any one antigenic site, but rather correlates with the angle of attachment and projection of the Fc above the surrounding canopy of F in the viral membrane (Figure 1 and 2). In polyclonal serum, antibodies with these characteristics may coexist, binding to non-overlapping epitopes and effectively enhancing complement activation by increasing the density of Fcs available for hexamer formation. In comparison, complement inhibition by polyclonal antibodies could occur if a subset of antibodies bind strongly to a productive antigenic site (*e.g*. site ), but in a non-productive orientation (*e.g*. with their Fc angled downward). It is unclear how common such antibodies may be, although it is worth noting that we are unaware of any examples among antibodies that bind near the apex of F. Based on this, we speculate that polyclonal responses are more likely to enhance than to inhibit the efficiency of IgG hexamer formation and activation of the classical pathway.

2) As quantitation of fluorescence intensities is crucial for this study, how did the authors ensure their detection was in a linear sensitivity range for the detectors for the various fluorescent channels? How is the "density of antigen" quantitated? What fraction of F or G is labeled?

We have performed additional characterization of our imaging and labeling approach to verify that the fluorescence intensities reported throughout the manuscript accurately reflect the concentration of different molecular species on RSV particles. This characterization is summarized below, and has been added to Figure 1 —figure supplements 2 and 3 in the revised manuscript. We have also updated the Methods section to describe how these measurements were performed (see Determining linearity of fluorescence measurements and Determining enzymatic labeling efficiencies for RSV F).

Determining density of antigen: All of the measurements in this work are reported as integrated fluorescent intensities. Since this reflects total abundance of antigen and not the density of antigen (*e.g*. antigen per unit area on the surface of a virus), we have revised our language to better reflect the data we are reporting. Specifically, we replace “antigen density” with “antigen abundance” where applicable.

Linearity of our measurements: To verify that our data collection falls within a linear regime, we used a D25 Fab^4^, labeled in a precise 1:1 (Fab:dye) ratio via a C-terminal ybbR tag attached to the heavy chain fragment. We chose to use the D25 Fab for several reasons: its ability to bind F protomers 1:1 at high concentrations; the accessibility of its epitope even on a crowded virion surface; and its exceptional affinity and slow dissociation. These characteristics allow us to image RSV particles with saturating amounts of bound D25 while still preserving low background fluorescence. We labeled these Fabs in two different colors using AF488 dye and sulfo-Cy5 (excited at 488nm and 640nm, respectively). Combining these labeled Fabs in defined ratios allows us to compare intensity values to what we would expect to see based on the input ratio of labeled Fab. This data, which is now included in Figure 1 —figure supplement 2,

demonstrates that fluorescence measurements obtained using identical illumination and acquisition settings as those used throughout the manuscript are linear across at least a ~250-fold range for absolute measurements (the maximum range that we tested), and a ~60,000-fold range for relative measurements. We also note that in selecting our imaging settings for the data in this work, we acted conservatively to avoid saturating our camera: across all wavelengths used (488nm, 560nm, and 640nm), our maximum pixel intensities fall within a range of 3000-10000 for a camera with 16-bit depth, <20% of the available dynamic range. Finally, we note that some images in the manuscript are displayed at high contrast, making them appear to be saturated. We have done this for the purposes of visualization, in an attempt to accommodate the large range of particle sizes and intensities present within a field of view. All quantification is performed on the raw images, which occupy a linear regime where intensities reliably reflect the amount of labeled protein associated with a virus.

Labeling efficiency: Determining labeling efficiencies for a single protein in a complex biological sample is technically challenging. To address this important question, we used a strategy analogous to our assessments of linearity to obtain an estimate of what fraction of F is fluorescently labeled by SrtA in our experiments. Although a similar approach could be taken for G, we have focused on F since all of our quantitative results are focused on this protein. The results of this analysis are presented in Figure 1 —figure supplement 3 in the revised manuscript. As before, we use the D25 Fab fragment, which binds with exceptional affinity to pre-F at up to three Fabs per F trimer^4^. We used the following procedure:

1. Label a first batch of viruses with equal amounts of D25(AF488) and D25(Cy5) Fab.

2. Image to determine the conversion factor relating the intensities of the two dyes when they are bound to RSV at a 1:1 ratio.

3. Label a second batch of viruses with Cy5 via SrtA and with saturating amounts of D25(AF488).

4. Calibrate the dye intensities for the second batch using the conversion factor obtained from the first batch to estimate the labeling efficiency via SrtA (i.e. the number of Cy5 molecules relative to the number of AF488 molecules).

Viruses simultaneously labeled with saturating, equimolar amounts of D25(AF488) and D25(Cy5) showed excellent linearity between the two dyes (Figure 1 —figure supplement 3A). Comparing the relative intensities gave a calibration constant of 3.14 and 3.90 for two biological replicates. This indicates that under our imaging conditions, AF488 is approximately ~3.5 times brighter than Cy5. We next proceeded to apply this calibration constant to viruses that were enzymatically labeled with Cy5 (via SrtA) and bound with saturating concentrations of D25 Fab labeled with AF488. This allows us to compare the abundance of enzymatically conjugated Cy5 with the abundance of D25 Fab per virus. This analysis suggests that the labeling ratio is >90% (Figure 1 —figure supplement 3B). We note that this is likely a modest over-estimate, since SrtA can conjugate fluorophores to both pre- and post-F, whereas D25 is specific to pre-F. However, our prior results suggest that the amount of post-F on these samples is modest, amounting to a few percent of total F (Figure 5).

To corroborate this analysis, we also directly compared intensities of viruses labeled with Cy5 via SrtA to viruses labeled with Cy5 via D25 Fab (Figure 1 —figure supplement 3C). This is less direct in that it requires separate virus samples, but more direct in the sense that it bypasses the need to calibrate dye intensities. This approach yields similar results, with the SrtA sample giving a median intensity that is 1.04 and 1.18 times brighter than the D25 Fab across two biological replicates, supporting the conclusion that SrtA labeling of RSV-F on the surface of virions is nearly complete. In these comparisons, values that are modestly greater than one could arise due to the specificity of D25 for pre-F, versus the ability of SrtA to label both pre- and post-F.

3) Instead of "complement activation," which involves much more than individual binding of complement components, it is better to use more specific terminology that accurately describes was actually measured (i.e. fluorescently labeled complement component binding).

We thank the reviewers for alerting us to this imprecise use of language. We have revised the manuscript to more accurately reflect what we are quantifying in our experiments, replacing “activation” with “deposition” where appropriate, including the title. We also use “activation of the classical pathway” in some instances, which has a narrower definition than “complement activation” and is more appropriate for our experiments.

4) The authors conclude that larger, globular particles are more susceptible to C3 deposition. The mechanistic link between size/globular morphology and post-F abundance, and the potential confounding effect of the latter on C3 deposition needs to be discussed in more detail.

The reviewers correctly point out that a number of variables influence the susceptibility of a virus particle to C3 deposition, including the availability of suitable epitopes for antibodies to bind and the geometry of the virus particle. In our revised manuscript, we have attempted to streamline our discussion of these points to improve clarity.

We have reorganized the *Results section* so that effects from particle morphology and the conversion of pre-F to post-F are discussed separately, in sections with more descriptive headings. Previously, these results had been discussed simultaneously, which we agree is confusing. We have also modified Figures 4 and 5 (Figures 3 and 4 in the original submission), so that Figure 4 focuses on our initial observations that globular morphology correlates with increased complement deposition, while Figure 5 demonstrates that this phenomenon is observed using either pre-F or post-F specific antibodies, even though the amount of pre-F is reduced on the surface of globular particles. We hypothesize that this is because the effects of decreased surface curvature are capable of compensating for the moderate loss of epitopes, and we proceed to investigate this hypothesis for the remainder of the *Results section*.

A related point raised by the reviewers is whether the link between globular morphology and post-F abundance is coincidental or has a mechanistic basis. We performed additional experiments to gain insight into this (Figure 6 —figure supplement 1B). Specifically, we compared the rate at which pre-F converts to post-F on particles that were preserved as filaments to those where globular morphology was induced by transient osmotic swelling. We find that both particle types have matching post-F to pre-F ratios initially, but that the globular particles have ~3.5-fold higher post-F to pre-F ratios following a 24h incubation. This suggests a mechanistic link, where changes in shape that arise from detachment of the viral matrix serve to destabilize the prefusion conformation of F. Finally, we have added a conceptual illustration to help clarify our working model regarding the relationship between RSV shape, sensitivity to complement deposition, and post-F abundance (Figure 6 —figure supplement 3).

5) Related to point 4), throughout the manuscript, the back and forth discussion on the relationship among virion morphology, post-F abundance, IgG concentration, and binding of complement components is convoluted. A streamlined description and/or illustration of the proposed working model(s) based on the data is needed.

We have revised our *Results section* to streamline the presentation of the multiple factors that influence complement deposition on RSV particles. This is discussed in more detail in our response to point (4) above. We have also added a conceptual illustration to summarize and clarify our working model (Figure 6 —figure supplement 3).

6) As a major conclusion of the work is that curvature of the virus surface has an outsize impact on the ability of complement proteins to bind, some effort at modeling this effect based upon known structures is warranted.

We agree with the reviewers’ comment that this is an important point that warrants further analysis and discussion. In response, we have updated the *Results section* to include a geometry-based model of how surfaces with different curvature differ in their ability to accommodate IgG1 hexamers. Consistent with our experimental observations, we find that surface curvatures in the range expected for RSV filaments have substantially less ability to accommodate antibody hexamers than do surfaces whose curvature is even just a few-fold lower. This model and its predictions are summarized in the newly added Figure 7. We also include our Matlab source code for anyone interested in further exploring the parameters of our simulations. A description of the model is reported in the *Methods* section (see *Modeling IgG hexamers on curved surfaces*).

7) The extent to which the results regarding virion morphology are biologically relevant needs to be better discussed, particularly with respect to design of vaccine candidates, including the failed formalin-inactivated RSV vaccine that was enriched in post-F.

We have revised the manuscript to clarify how our results may relate to more physiological in vivo settings. We have also revised the *Discussion section* (paragraph 5) to address potential connections to the formalin-inactivated vaccine trial. Our revisions and the reasoning behind them are summarized below.

Biological relevance of changes in virion morphology: RSV particles are notoriously malleable, and it is well established that purification and biochemical treatments can alter morphology. Although we use this sensitivity to change particle characteristics in a number of our experiments, we also perform experiments in which the particles are not subjected to any harsh treatment, but rather allowed to age at physiological temperatures (Figure 5). The fact that these samples nonetheless show a significant increase in the percentage of globular particles suggests that handling or harsh treatment is not necessary to disrupt the structure of RSV. In support of this idea, we have also imaged RSV on the surface of cells as it spreads during infection (Figure 5 —figure supplement 1). These images reveal the presence of globular particles in the complete absence of handling beyond the gentle dilution of pre- and post-F antibodies directly into the culture media. Importantly, our work does not demonstrate that the same phenomenon occurs in vivo. We acknowledge this caveat in our discussion (Line 401) by writing:

“Although more work is needed to understand the prevalence of globular particles during in vivo infection, our observations that these particles accumulate over time through the conversion of viral filaments – even under normal cell culture conditions – suggest that their presence in vivo is feasible, where the physical and chemical environment would be considerably harsher and more complex.”

Relevance to past and present vaccine candidates: The sensitivity of RSV to any mechanical perturbation and its tendency to change morphology spontaneously over time would lead us to expect that the formalin-inactivated vaccine would show the same tendency. This vaccine was prepared from cell culture supernatants that were filtered, concentrated by centrifugation, inactivated with formalin, and adsorbed to alum^5,6^. Although we have not attempted to reproduce this preparation, it is likely that this treatment would result in the loss of filamentous morphology, and presumably an increase in complement deposition on the inactivated particles. Given the prominent and wide-ranging effects of complement on humoral immunity, this may generate a different response than if RSV antigens were administered as filamentous particles. This connection is now discussed in paragraph 5 of the revised *Discussion*.

Reviewer #1 (Recommendations for the authors):To address minor weaknesses above, a schematic to illustrate the working model(s) would be helpful.

We have added an illustration of our working model in Figure 6 – figure supplement 3. In this illustration, we haveattempted to highlight how changes in RSV morphology driven by detachment of the viral matrix leads to increasedcomplement deposition and – separately - to increased abundance of post-F. We speculate that these collective changesin the molecules / epitopes presented on the viral surface may shape downstream immune responses to RSV infection orvaccination (including the possibilities discussed in paragraphs 4 and 5 of the *Discussion*).

Reviewer #2 (Recommendations for the authors):1) I appreciate the clarity of the figures!

Thank you – we have attempted to improve the clarity in other aspects of the manuscript as well.

2) Error bars in Figure 2 C, D. Figure 2C caption please clarify: "Connecting lines show data from paired biological replicates". Does the "paired replicates" here refer to the experiment performed two independent times from separate virus preparations? Or as in other experiments 3 biological replicates were performed? Or does the pair refer to the paired experiments such as +/-CD55? In Figures 4 and 5 this was done, but here it is not clear. Please include error bars as well.

Thank you for pointing this out. Connecting lines in Figure 2C and 2D (Figure 3C and 3D in the revised manuscript) indicate samples that were prepared and imaged in parallel. These samples are not biologically paired as we incorrectly stated because they necessarily come from separate batches of infected cells (wildtype and CD55 knockout). Nonetheless, because these samples were prepared and imaged on the same day, using a common batch of complement components, we feel that it is appropriate to treat them as paired replicates.

The revised caption for Figure 3C now reads:

(C) Comparison of C3 deposition on viruses raised in wildtype cells (open circles) or CD55 knockout cells (closed circles) using three different F-specific antibodies and a negative control. Black markers show average values across four biological replicates. Individual replicates are shown in gray. Connecting lines indicate samples that were prepared in parallel, using virus collected from separate batches of wildtype and CD55 knockout cells. P-values are determined using a paired-sample t-test.

Rather than displaying error bars for this data, we show results for individual replicates (gray) along with the average results across all replicates (black). We feel that this conveys the same information as error bars in a more direct way.

3) Figure 5D the grey individual replicates are so faint as to be almost invisible.

We have increased the contrast of these lines to make them more visible.

4) Some typos here and there. E.g. Figure 1 caption "positive", line 340 "asses".

Thank you for calling our attention to these typos – we have corrected these and any others that we were able to find.

Reviewer #3 (Recommendations for the authors):1. The authors may want to include mention of the formalin-inactivated RSV vaccine candidate that caused vaccine-enhanced disease in a clinical trial that resulted in the deaths of 2 infants. Killikelly and Graham (Sci Rep. 2016 Sep 29;6:34108) showed that the formalin treatment caused all of the prefusion F to convert to postfusion F, and although not investigated in that study, the virion purification and incubation with formalin likely converted all particles to the pleiomorphic/spherical form.

Thank you for this suggestion. We agree with the reviewer’s interpretation, and have added this point to paragraph 5 of the discussion.

2. The schematic in Figure 1A is incorrectly drawn since the 3D3 antibody that captures the virion to the coverslips should be binding to G and not F.

Thank you for catching this oversight – the schematic in Figure 1A has been updated to illustrate our particle immobilization strategy more accurately.

References

1. Lavinder, J. J. *et al.,* Identification and characterization of the constituent human serum antibodies elicited by vaccination. *Proc Natl Acad Sci USA* 111, 2259–2264 (2014).

2. Demonbreun, A. R. *et al.,* Comparison of IgG and neutralizing antibody responses after one or two doses of COVID-19 mRNA vaccine in previously infected and uninfected individuals. *EClinicalMedicine* 38, 101018 (2021).

3. Lee, J. M. *et al.,* Mapping person-to-person variation in viral mutations that escape polyclonal serum targeting influenza hemagglutinin. *eLife*
**,**, e49324 (2019).

4. McLellan, J. S. *et al.,* Structure of RSV fusion glycoprotein trimer bound to a prefusion-specific neutralizing antibody. *Science* 340, 1113–1117 (2013).

5. Fulginiti, V. A. *et al.,* RESPIRATORY VIRUS IMMUNIZATION. *American Journal of Epidemiology* 89, 435–448 (1969).

6. Kim, H. W. *et al.,* Respiratory syncytial virus disease in infants despite prior administration of antigenic inactivated vaccine. *Am J Epidemiol* 89, 422–434 (1969).

7. Needham, D. and Nunn, R. S. Elastic deformation and failure of lipid bilayer membranes containing cholesterol. *Biophysical Journal* 58, 997–1009 (1990).

8. Sharp, T. H. *et al.,* Insights into IgM-mediated complement activation based on in situ structures of IgM-C1-C4b. *Proc Natl Acad Sci USA* 201901841 (2019) doi:10.1073/pnas.1901841116.

9. Tradtrantip, L., Yao, X., Su, T., Smith, A. J. and Verkman, A. S. Bystander mechanism for complement-initiated early oligodendrocyte injury in neuromyelitis optica. *Acta Neuropathol* 134, 35–44 (2017).

10. Marcandalli, J. *et al.,* Induction of Potent Neutralizing Antibody Responses by a Designed Protein Nanoparticle Vaccine for Respiratory Syncytial Virus. *Cell* 176, 1420-1431.e17 (2019).